**Subject Category:**
Biology (whole organism)

taxonomy and systematics/palaeontology/biogeography

*Muntiacus gigas*, *vuquangensis*, zooarchaeology, Vietnam, Annamites

**Author for correspondence:**
C. M. Stimpson
e-mail: c.stimpson@qub.ac.uk

# An 11 000-year-old giant muntjac subfossil from Northern Vietnam: implications for past and present populations

C. M. Stimpson[1], B. Utting[2], S. O'Donnell[1],
N. T. M. Huong[3], T. Kahlert[1], B. V. Manh[4], P. S. Khanh[5]
and R. J. Rabett[1]

[1]School of Natural and Built Environment, Queen's University Belfast, Elmwood Avenue, Belfast BT7 1NN, UK
[2]Department of Archaeology, University of Cambridge, Downing Street, Cambridge CB2 3DZ, UK
[3]Vietnam Academy of Social Sciences, Institute of Archaeology, 61 Phan Chu Trinh Str., Hoan Kiem, Hanoi, Vietnam
[4]Department of Tourism, No 06, Tràng An Street, Đông Thành ward, Ninh Bình city, Ninh Bình province, Vietnam
[5]Tràng An Landscape Complex Management Board, Ninh Bình City, Vietnam

CMS, 0000-0003-4327-4987; SO, 0000-0003-0731-7425

Described at the end of the twentieth century, the large-antlered or giant muntjac, *Muntiacus gigas* (syn. *vuquangensis*), is a Critically Endangered species currently restricted to the Annamite region in Southeast Asia. Here we report subfossil evidence of giant muntjac, a mandible fragment dated between 11.1 and 11.4 thousand years before present, from northern Vietnam. We describe morphological and metric criteria for diagnosis and consider the specimen in the context of regional archaeological and palaeontological records of *Muntiacus*. We then consider the palaeoenvironmental context of the specimen and the implications for habitat requirements for extant populations. The new specimen extends the known spatial and temporal range of giant muntjacs in Vietnam and is further evidence that this species was more widely distributed in the Holocene than current records indicate. While regional proxy evidence indicates a drier climate and more open woodland habitats at the onset of the Holocene, contextual evidence indicates that the specimen derived from an animal inhabiting limestone karst forest. This record also supports the assertion that remnant populations are in a refugial state, as a result of anthropogenic pressures, rather

than representing a centre of endemism. These facts underscore the urgent need for the conservation of remaining populations.

# 1. Introduction

Human activities continue to reduce mammal populations, geographical ranges and, ultimately, cause extinctions [1]. While the scale, rapidity and mechanisms driving recent anthropogenic impacts such as these are unprecedented [2,3], attempts to form effective conservation strategies are potentially hampered by a paucity of studies that consider biological communities from millennial as well as ecological (typically less than 50 years) timescales [2,4–7]. Current studies and recent data (less than 100 years) characterize animal communities and their habitats only after, potentially, centuries or millennia of exploitation and modification by humans [7–10]. In this sense, a historical amnesia results in a shifting baseline syndrome where ecosystems, animal populations and their current geographical distribution are interpreted as 'natural' or pristine, when they are in fact degraded [2,9,11]. This issue is likely to be particularly problematic with rare, recently described and poorly-known mammals [7,11]. In this context, the potential of Quaternary archaeological and palaeontological data to provide longer time-scale perspectives and benchmark evidence for biological conservation is increasingly being recognized and demonstrated [7,9–16]. Here, we present just such a line of evidence and consider a poorly-known and Critically Endangered species of deer (Cervidae): the giant muntjac.

Muntjacs (Cervidae, *Muntiacus*) are a taxonomically diverse group of small solitary deer distributed throughout South, Southeast and East Asia. Over a dozen extant species have been proposed [17] of which many were scientifically described only recently, at the close of the twentieth century [18–22]. While the taxonomic status of several species remains a matter of debate [17,23,24], an unequivocally novel, large-bodied muntjac was reported in 1994 [22], one of a suite of new mammalian species to be described from the Annamite (Truong Son) region [18] of continental Southeast Asia (figure 1).

The new species, the large-antlered or giant muntjac (named as *Muntiacus vuquangensis*), was originally assigned to a novel genus *Megamuntiacus* [22] although subsequent studies of morphology [27] and molecular phylogenetics [28] confirmed its placement within *Muntiacus*, with a close affinity to Reeve's (*M. reevesi*) and the so-called 'little' muntjacs (*M. puhoatensis*, *M. truongsonensis* and *M. rooseveltorum*) [29,30]. The giant muntjac resembles its congenerics but is distinguished by shorter, thicker pedicles and characteristically larger, robust antlers with prominent longitudinal furrows and well-developed brow tine [7,27]. Indeed, much of the extant (or recent) range of the giant muntjac has been interpolated from the presence of hunting trophies consisting of frontlets and antlers [31]. The giant muntjac is also relatively large in terms of overall stature and body mass within the genus. Although large individual 'red' (*M. muntjak*) and 'black' (*M. crinifrons*) muntjacs are likely to overlap with the lower size and weight range of the species, giant muntjacs are known to exceed 30 kg in weight [27] and are likely to range up to 50 kg [32]. The giant muntjac is known only from the Annamite region of Lao PDR, Vietnam and eastern Cambodia and is classified by the IUCN as Critically Endangered with a high probability of extinction in less than 20 years [33].

Four years prior to the discovery and description of *M. vuquangensis*, however, Wei *et al.* [34] described a novel (and at the time, thought to be extinct) large muntjac species, *Muntiacus gigas*, based on examination of subfossil antlers from the Chinese Neolithic site of Hemudu in the Yangtze delta, dated 6 to 7 ka ('ka' = thousands of years before present). Recent work by Turvey *et al.* [7] on the morphometric characteristics of these specimens, with further samples from several Chinese sites ranging in date from the Late Pleistocene to Holocene (figure 1; sites 1–8), demonstrated that there are no morphological grounds to separate *M. gigas* specimens from extant *M. vuquangensis* and that these taxa should be considered conspecific. Given that the description of *M. gigas* predated the description of *M. vuquangensis*, the species name of the former has priority [7]. We therefore adopt the use of *M. gigas*, which we consider to be synonymous with *M. vuquangensis*.

Here, we report the identification of a large *Muntiacus* mandible from an archaeological context in northern Vietnam, which we refer to *M. gigas*. The specimen was recovered during excavations of an anthropogenic shell midden, which dates from the Late Pleistocene 13.6 ka to the beginning of the Holocene 10.5 ka [35], from the cave site of Hang Boi in the Tràng An World Heritage Area (figure 1). We first describe the morphological and metric criteria for our diagnosis of the specimen. We then briefly review palaeontological and archaeological records of *Muntiacus* dental remains from the

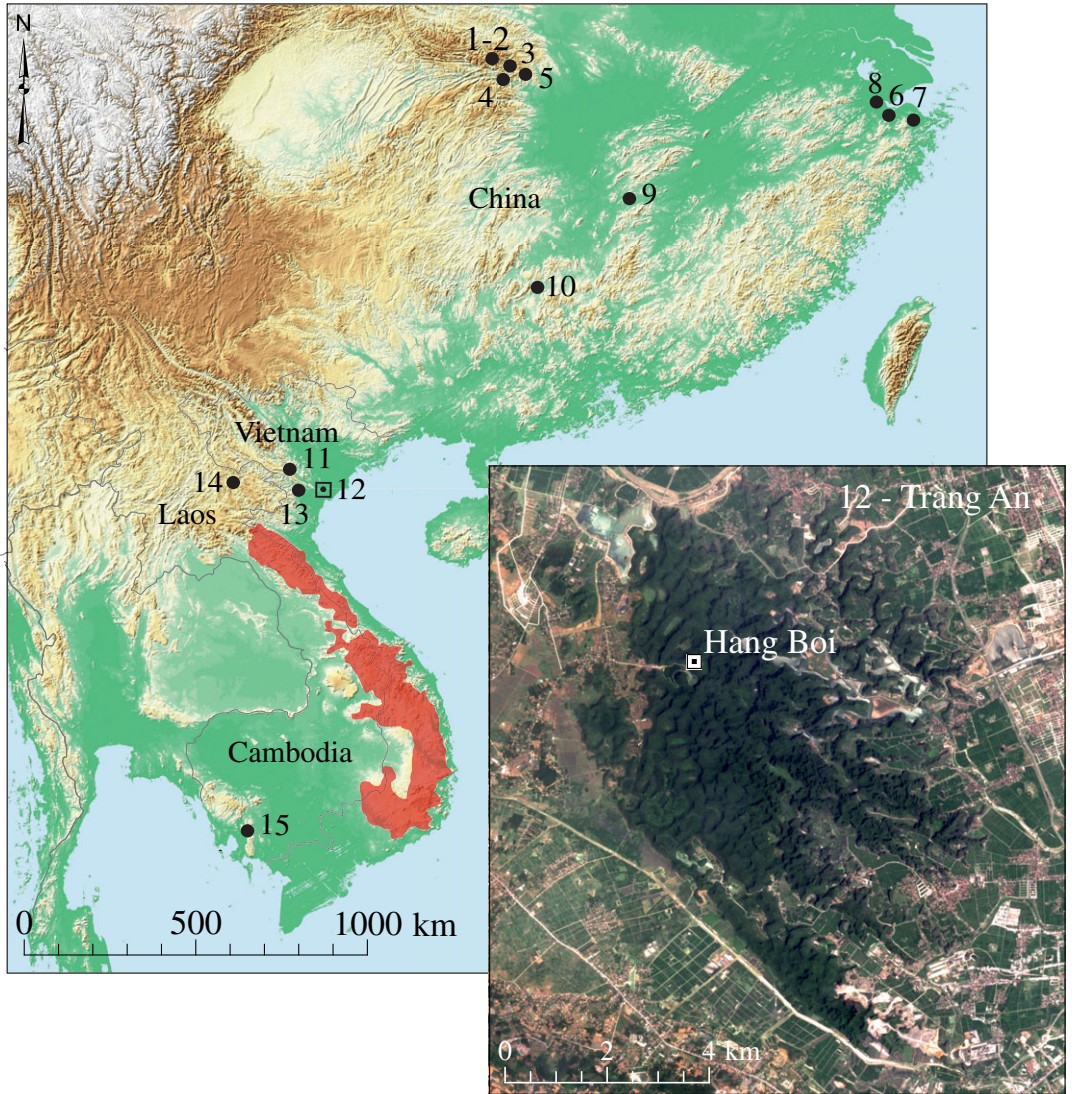

**Figure 1.** East Asia, showing the current distribution of *M. vuquangensis* [16] in the Annamite region (red shading), selected archaeological and palaeontological sites with *Muntiacus* remains mentioned in the text and location of Hang Boi (site 12) in the limestone karst of the Tràng An World Heritage Area (inset). Holocene sites—China: 1, Guanzhuangping; 2, Liulinxi; 3, Xisiping; 4, Luoping; 5, Lujiahe; 6, Tianluoshan; 7, Hemudu [7]. Late Pleistocene sites—China: 8, Yuhang; 9 [7], Yanjiawan caves; 10, Fuyan cave [25]. Late Pleistocene—Vietnam: 11, Duoi U'Oi; 13, Lang Trang. Middle Pleistocene sites—Laos and Cambodia: 14, Tam Hang; 15, Phnom Loang [26].

region and assess the evidence for the possible presence of giant muntjac in other records. Finally, we discuss the implications of this new find for remaining extant populations.

## 2. Material and methods

### 2.1. Location

The Tràng An World Heritage Area (TAWHA) is located near the southern margin of the Red River delta, approximately 90 km southwest of Hanoi, in Ninh Binh Province, northern Vietnam (figure 1). The property covers 6226 hectares of the Tràng An karst, an isolated massif of highly-dissected Triassic limestone rising from a low lying plain that extends to the present coastline of the South China Sea, approximately 35 km to the south east. The outcrops of the massif are vegetated by highly adapted, edaphic limestone karst forest. The core property is surrounded by a buffer zone of 6026 hectares of mostly rural land with rice paddy fields, local communes and a large pagoda complex. Tràng An was inscribed on the World Heritage List as a mixed Cultural and Natural property in June 2014.

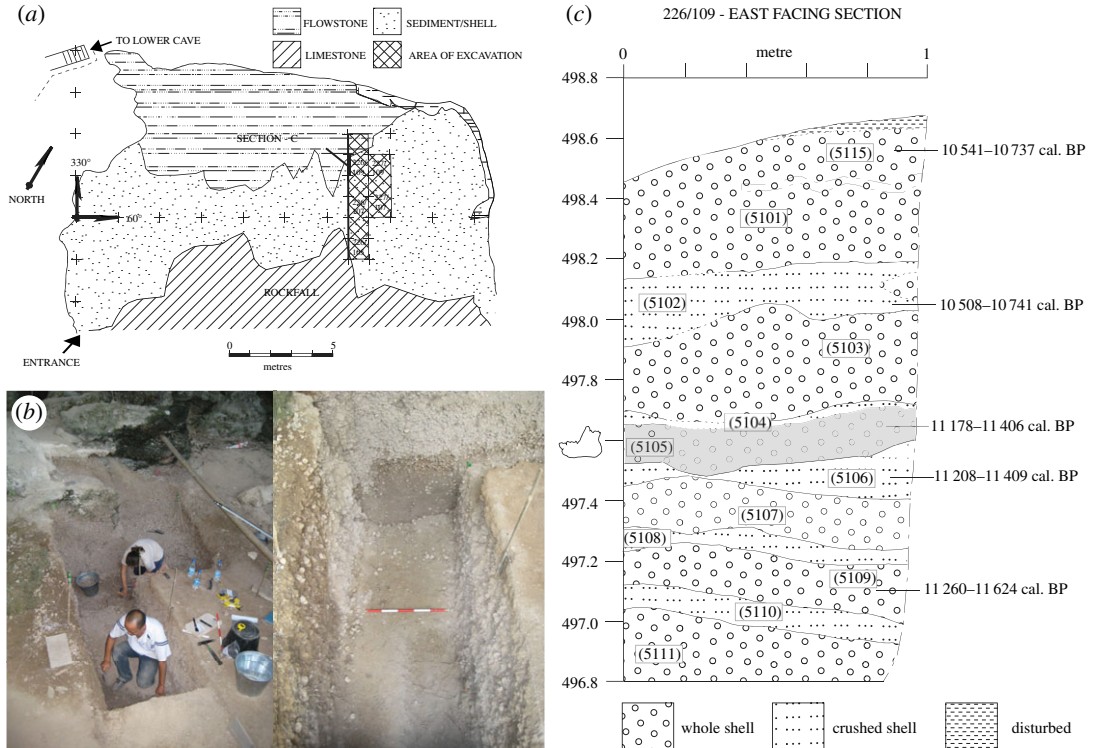

**Figure 2.** Hang Boi. (*a*) Plan of cave showing excavation area: grid square 226/109 and section are indicated (*b*) looking NNW/NW during excavation through shell midden, and (*c*) Upper 2 m of stratigraphic section (east facing) of grid square 226/109 showing calibrated radiocarbon dates from charcoal and context (5105) containing specimen HBC-27587. Heights shown are relative to arbitrary site datum set at 500 m.

Hang Boi (figure 2) was the subject of archaeological investigations from 2007 to 2010 by the Tràng An Archaeological Project (TAAP) and the geographical, geological and archaeological contexts of the cave have been described in detail [35–37]. In brief, the caverns of Hang Boi (20°15′32″ N; 105°53′17″ E) are located near the apex of a karst outcrop, approximately 78 m above sea level, in the north of the massif in the core of the property (figure 1—inset). The cave entrance faces SSE and overlooks a small doline (approx. 3 m asl). Archaeological excavations focused on shell-rich midden deposits that had accumulated in the upper cave entrance of Hang Boi (figure 2).

## 2.2. Stratigraphic and chronological setting

The mandible, specimen number HBC-27587, was recovered from a relatively deep (approx. 3 m) but chronologically-constrained shell midden sequence (figure 2). The principal mechanism of accumulation was human subsistence activities [35–37] and the mandible is most parsimoniously a remnant of an animal that had been consumed. Shells of *Cyclophorus* spp. form the main component of the midden with a matrix of friable silty clay, hardened in places by calcium carbonate deposition from cave driplines, the remains of broken and occasionally burnt animal bone from a range of vertebrate taxa (including Cervidae, Bovidae, Primates, Carnivora, Testudines and Aves) and numerous fragments of charcoal [35]. The stratigraphy of the midden is complex but alternating layers of crushed shell—former surface areas subjected to trampling—and whole shell layers, attributed to discard episodes, are detectable in section (figure 2). The mandible was recovered in the western half of a 1 m × 1 m grid square (226/109) in a 10 cm unit of excavation (5010) through context (5105), just over 1 m below the maximum height of the preserved surface of the midden (figure 2). This deposit derived from the latest and apparently most intensive phase of human occupation revealed in the excavated sequence and is dated to the early Holocene [35]. While limited root intrusion was very occasionally observed in the deposit, there was no indication of significant bioturbation or disturbance, and no evidence to suggest that the specimen is intrusive or otherwise *ex situ*.

AMS radiocarbon dates have been derived from charcoal samples that were recovered throughout the midden sequence [35,36] and calibrated using the Intcal. 13 calibration curve [38]. Following convention,

the beginning of the Holocene is defined as 11 650 years ago [39]. Calibrated radiocarbon dates are shown here as two sigma ranges as 'cal. BP' ('calibrated years before present', where 'present' is defined as 1950). Calibrated dates from the whole midden sequence are in superposition and indicate that the excavated deposits accumulated between the end of the Pleistocene, 13.6 ka, to early Holocene, 10.5 ka (electronic supplementary material, table S1). A charcoal sample (UBA-8372) recovered from the same excavation unit as the mandible (5010, in context 5105) yielded a calibrated radiocarbon date range of 11 178–11 406 cal. BP (figure 2).

## 2.3. Analyses: morphology, metrics and conventions

The identification of the mandible was made as part of the ongoing review of the vertebrate remains from the TAWHA recovered by the TAAP and the subsequent SUNDASIA project. Analysis was carried out in the UK at the Oxford University Museum of Natural History with the permission of the Tràng An Management Board and Ninh Binh Peoples Committee. The specimen is to be stored and curated by the Tràng An Management Board, Ninh Binh.

Dental terminology follows Bärmann & Rössner [40]. Mandible and dental characters referred to in the text are summarized in figure 3. Measurements of the mandible and lower molars follow the conventions described in Janis ([42]; measurements are annotated in electronic supplementary material, figure S1). Measurements were taken with dial callipers to the nearest 0.01 mm (table 1). Body mass estimates were derived from six measurements of the specimen, following the least-squares regression equations and percentage standard errors of estimate (%SEE) for the Cervidae of Janis ([42]; table 2).

The mandible was identified by morphological and metric comparisons with comparative museum specimens from regional Cervidae and Bovidae, with published descriptions and with morphometric data. Modern comparative specimens were consulted at the Oxford University Museum of Natural History—OUMNH, University Museum of Zoology, Cambridge—UMZC, American Museum of Natural History—AMNH and Natural History Museum UK—NHMUK (electronic supplementary material, table S2). Wherever possible, reference specimens with equivalent individual dental age stages (IDAS) [41] to the archaeological specimen (IDAS 3 to 4) were prioritized for morphological comparisons of the dentition.

Statistical tests were performed to compare dental metrics of comparative museum specimens and published sources [25,26,43]. For comparison of dental metrics from the mandible, and in cases where sample sizes of comparative data were small, testing followed the procedures for normative comparisons [44–47], where metrics from a single case (e.g. HBC-27587) are compared with a normative sample generated from comparative data to test for differences from the estimated parameters of a given taxon (e.g. *M. muntjak*). Multivariate normative comparisons (m2 and m3, lengths and widths or a subset of these data depending on the availability of comparative data) were performed using a modified Hotelling's $T^2$ test, following Huizenga *et al.* [46]. Univariate normative comparisons were also performed with a modified *t*-test [44,45] with a step-down correction to control familywise false-positive error rate in multiple comparisons [47]. These approaches assume univariate/multivariate normality in the normative samples. Sample datasets were examined for departures from normality using the functions in PAST 3.20. Normative comparisons were performed using the E-clip, Multivariate and Univariate Normative Comparisons online platform [48] accessed at: eclip.shinyapps.io/NormativeComparisons/.

Data from archaeological and palaeontological records of *Muntiacus* spp. from Pleistocene sites in the region were also drawn from published sources [25,26]. A limited examination of trends in lengths of the third lower molar (which are more readily identifiable to tooth position than other lower molars in the case of individual teeth and less prone to variance in length as a function of wear [49]) was undertaken to investigate the possible presence of giant muntjacs in deeper time records. Given that samples may incorporate teeth from two or more different *Muntiacus* species, with possibly different distributions of measurements, these data were analysed with non-parametric statistics in PAST 3.20.

# 3. Results

## 3.1. Morphology—description

Specimen HBC-27587 (figure 4) is a fragment of a left mandible (greatest length of specimen = 78.86 mm; greatest height = 55.60 mm). Overall, the mandible appears gracile though the remnant of the body of the

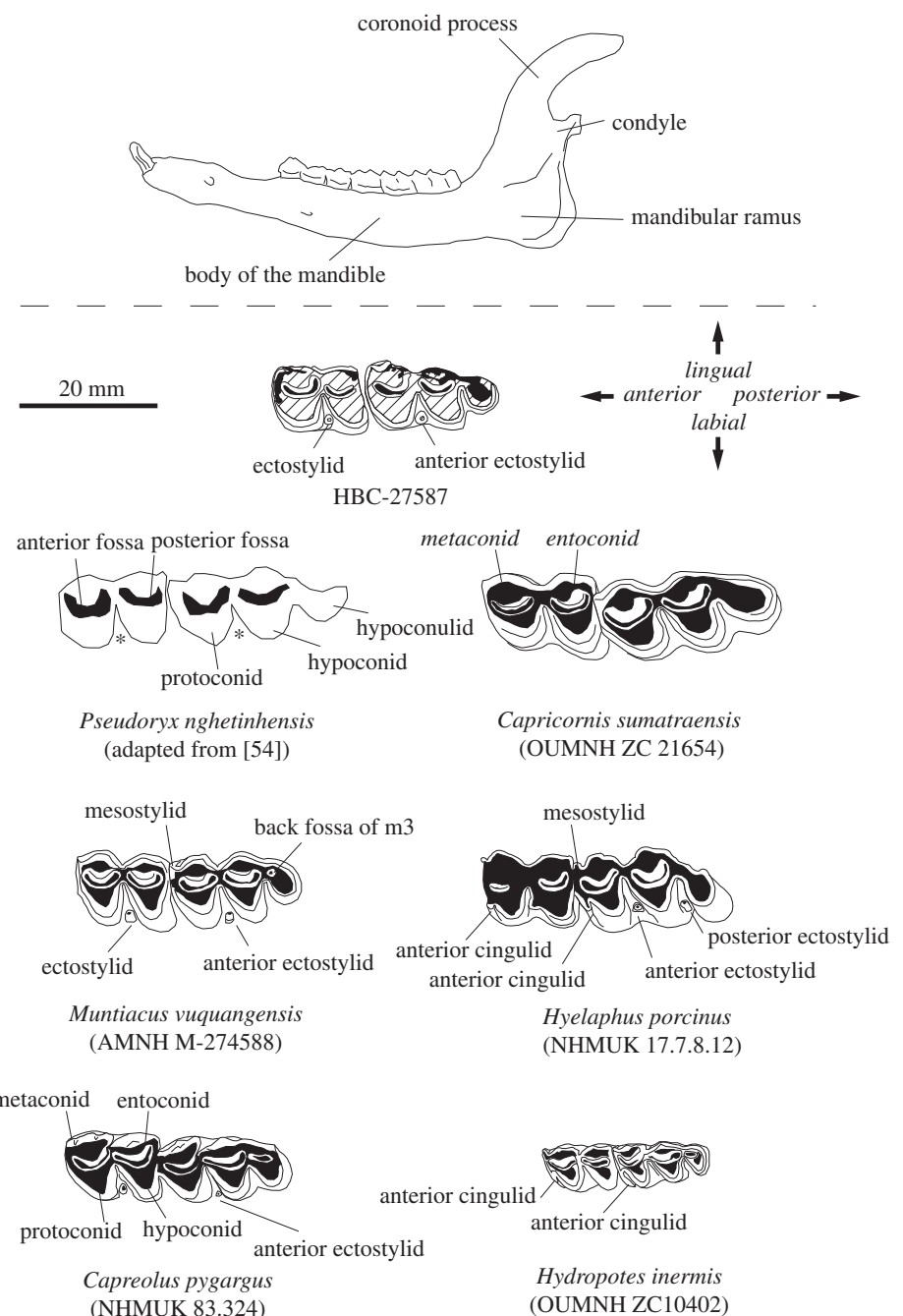

**Figure 3.** Annotated characters of the mandible (top) and second and third lower molars (bottom) referred to in the text. Molars of two species of Bovidae (*Pseudoryx nghetinhensis* and *Capricornis* [*sumatraensis*] *maritimus*) and four species of the Cervidae (*Muntiacus vuquangensis*, *Hyelaphus porcinus*, *Hydropotes inermis* and *Capreolus pygargus*) are shown to scale with specimen HBC-27587. All specimens are equivalent to individual dental age stages (IDAS) late stage 3 to stage 4 [41].

mandible is broken below the anterior roots of the second lower molar and has been compressed and crushed. The mandibular ramus is largely intact. The posterior edge of the ramus, between the angle and the remaining portion of the condyle, is concave. The articular facet of the condyloid process is not preserved. The coronoid process is broken just above (i.e. dorsally to) the level of the condyle.

Silty clay sediment adheres as a hard crust (consolidated by calcium carbonate deposition) to approximately 60% of the surface area of the specimen. This crust required manual removal from the occlusal, lingual and labial aspects of the teeth to examine their characteristics. A large and complete shell (*Cyclophorus* sp.) is adhered to the lingual face of the ramus, overlying the mandibular foramen (figure 4). The exposed bone on the remainder of the specimen is yellow/brown in colour.

The second (m2) and third (m3) lower molars are *in situ* and in wear (equivalent to IDAS late stage 3 or 4), indicating an adult animal (figure 4). The enamel is broken on the lingual margins of both teeth.

**Table 1.** Measurements of mandible HBC-27587. Measurements follow the conventions in Janis ([42]; electronic supplementary material, figure S1).

| measurement | definition | mm |
| --- | --- | --- |
| PJL | posterior jaw length | 43.46 mm |
| WMA | maximum width of mandibular angle (ramus) | 46.80 mm |
| SLML | length m2 | 14.01 mm |
| SLMW | width m2 | 9.42 mm |
| SLMA | area m2 | 11.49 mm$^2$ |
| TLML | length m3 | 19.67 mm |
| TLMW | width m3 | 9.97 mm |
| TLMA | area m3 | 14.00 mm$^2$ |

**Table 2.** Body mass estimates (est. kg = estimated kilograms) from six measurements of mandible HBC-27587. Equations follow Janis [42] for the Cervidae. $r^2$ values give an indication (but is not a measure of significance) of the variability of the dependant variable (body mass) collectively explained by the independent variables (jaw/teeth measurements) - over 93% (0.93) in all cases. %PE indicates the percent difference between actual weight and the estimated values in the samples used by Janis [42]. %SEE (percent standard error of the estimate) indicate the $\pm$ percent deviation from the estimate (illustrated in figure 6) in which 68% of actual values would be expected to fall.

| | mm | cm | log10 | $r^2$ | intercept | slope | %SEE | %PE | log10 kg | est. kg |
| --- | --- | --- | --- | --- | --- | --- | --- | --- | --- | --- |
| PJL | 43.46 | 4.346 | 0.63809 | 0.937 | 0.029 | 2.431 | 33.7 | 20.3 | 1.580196 | 38.03611 |
| WMA | 46.8 | 4.68 | 0.670246 | 0.949 | −0.53 | 3.181 | 30 | 19.4 | 1.602052 | 39.99927 |
| SLML | 14.01 | 1.401 | 0.146438 | 0.951 | 1.119 | 3.106 | 29.1 | 20.4 | 1.573837 | 37.48322 |
| SLMA | 11.49 | 1.149 | 0.060245 | 0.955 | 1.474 | 1.638 | 27.9 | 23 | 1.572681 | 37.38355 |
| TLML | 19.67 | 1.967 | 0.293804 | 0.957 | 0.799 | 3.143 | 27.4 | 19.1 | 1.722427 | 52.77486 |
| TLMA | 14.00 | 1.400 | 0.14625 | 0.953 | 1.346 | 1.561 | 28.8 | 19.9 | 1.574296 | 37.52285 |

The labial margins of the teeth are intact. On the m2, a worn but robust ectostylid is located between the protoconid and hypoconid. The anterior margin of the m2 has lost enamel at the point of contact with the posterior margin of the m1 (not present). No anterior cingulum is evident. The posterior margin of the m2 has lost enamel at the contact point with the anterior margin of the m3. Enamel is present on the labial margin of the hypoconid, though the entoconid is broken at the location of the metastylid. Enamel has broken away along the remainder of the lingual edge of the metaconid. The anterior and posterior fossae form closed loops. From the lingual aspect, the anterior root is broken. The posterior root is partially exposed in the broken mandibular body and appears intact. The occlusal surface of both teeth, when viewed from the anterior aspect, is concave.

The labial margins of the protoconid, hypoconid and hypoconulid of the m3 are preserved. A worn but well-developed anterior ectostylid is present between the protoconid and hypoconid. The preserved labial portion of the hypoconulid is rounded and appears to be robust and well-developed. The back fossa is lost through wear. The valley between the hypoconid and hypoconulid is narrow and orientated in the labial/lingual direction. The lingual margins of the entoconulid and entoconid are broken. Enamel remains on the anterior portion of the metaconid and the mesostylid. No anterior cingulum is evident.

## 3.2. Morphology and metrics—comparisons

Absolute size, lack of anterior cingula and tooth dimensions all indicate against the much smaller Moschidae (musk deer) (see also [25]). Mandibles of the Bovidae tend to be proportionally more robust than the archaeological specimen, with a much straighter posterior margin. Furthermore, the remaining lower molars in the archaeological specimen, though worn, appear to be rather low-crowned (brachydont) for the state of wear. The simple and rounded occlusal morphology (if not

**Figure 4.** (a) Lateral views of the left mandible of comparative specimens of male *Muntiacus reevesi* (OUMNH ZC SR0483) and male *M. muntjak* (OUMNH ZC 20196) shown to scale with specimen HBC-27587. (b–e) Specimen HBC-27587 shown in lateral (b), medial (c), dorsal (d) views and occlusal surfaces of m2 and m3 (e). All scale bars = 20 mm.

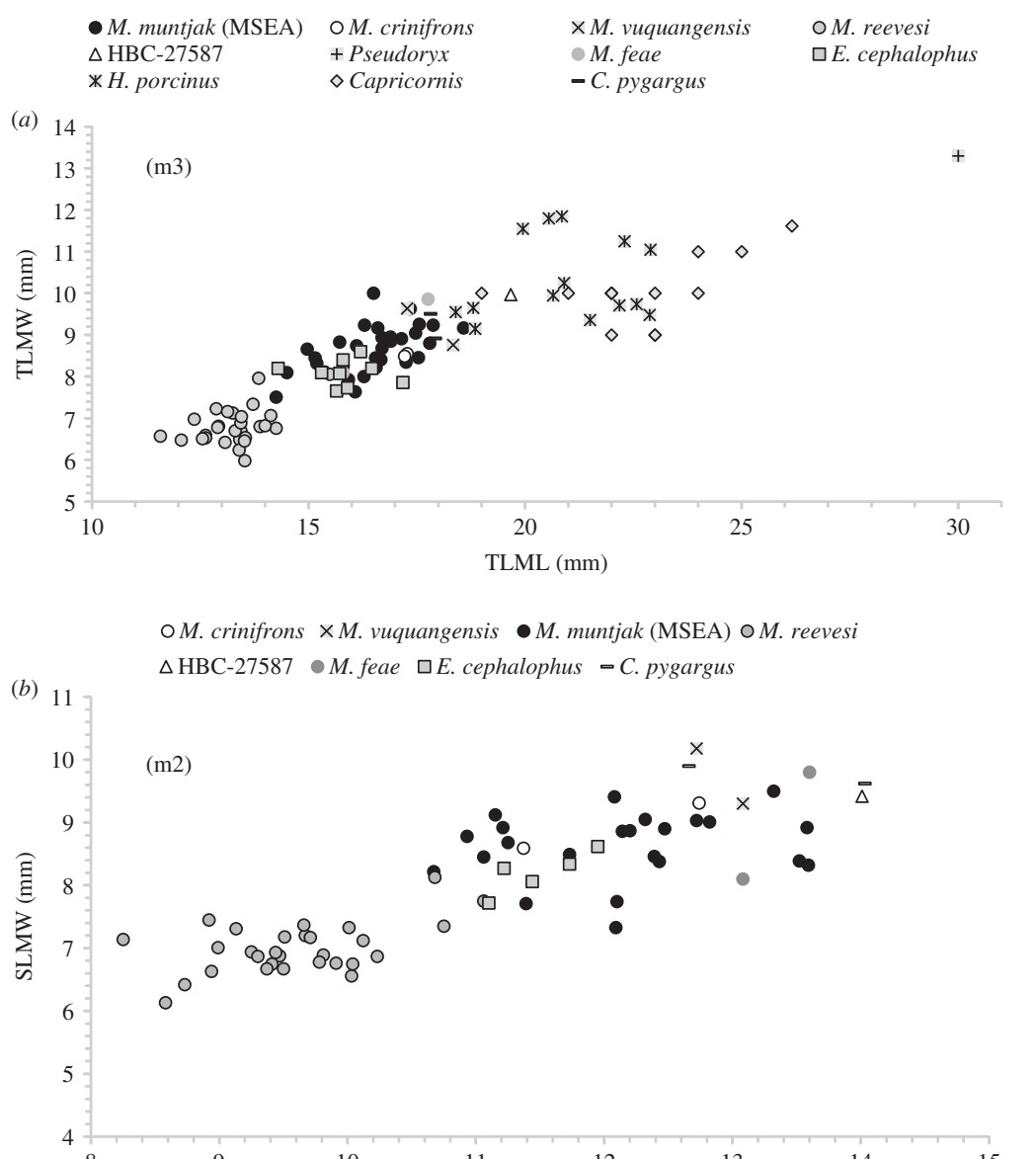

**Figure 5.** Bivariate plots to show, (*a*) length (TLML) and width (TLMW) measurements of the m3 from HBC-27587 in comparison to measurements from regional bovids and cervids and, (*b*) length (SLML) and width (SLMW) measurements of the m2 from HBC-27587 in comparison to measurements from the Munticiani and *Capreolus pygargus*. Data from comparative specimens (electronic supplementary material, table S2) and published sources [25,26,49–51].

crown height) of the teeth do resemble serow (genus *Capricornis*) in a similar state of wear (figure 3) and dental metrics indicate an overlap, in the lower size range, for equivalent data for *Capricornis* sp. (figure 5). Regional caprines (*Capricornis*, *Naemorhedus*) can be discounted, however, by the presence of robust ectostylids on the molars of the archaeological specimen. Developed ectostylids are not present on the second and third lower molars of *Capricornis* and *Naemorhedus* ([52]; see figure 3). In *Capricornis*, if cases of the presence of an ectostylid have been recorded, they are a rare and poorly developed character restricted to the m1 [49]. Conversely, examination of the dentition of the saola (Bovini—*Pseudoryx nghetinhensis*), first described from the Annamites in 1992 [53], has indicated that ectostylids are present on the lower molars and are regarded as one of several 'primitive' characters of this animal [54]. Available data are limited but saola are clearly much larger animals, reported to weigh in the region of 70–100 kg [55], with much more robust mandibles with a straight posterior margin of the ramus. Tooth measurements indicate that the lower molars are also much larger and more robust than the archaeological specimen (figure 5).

The morphology of the remnant mandible and preserved dentition are consistent with the Cervidae. Body weight estimates derived from six measurements (table 2) suggest a mid-sized deer, ranging

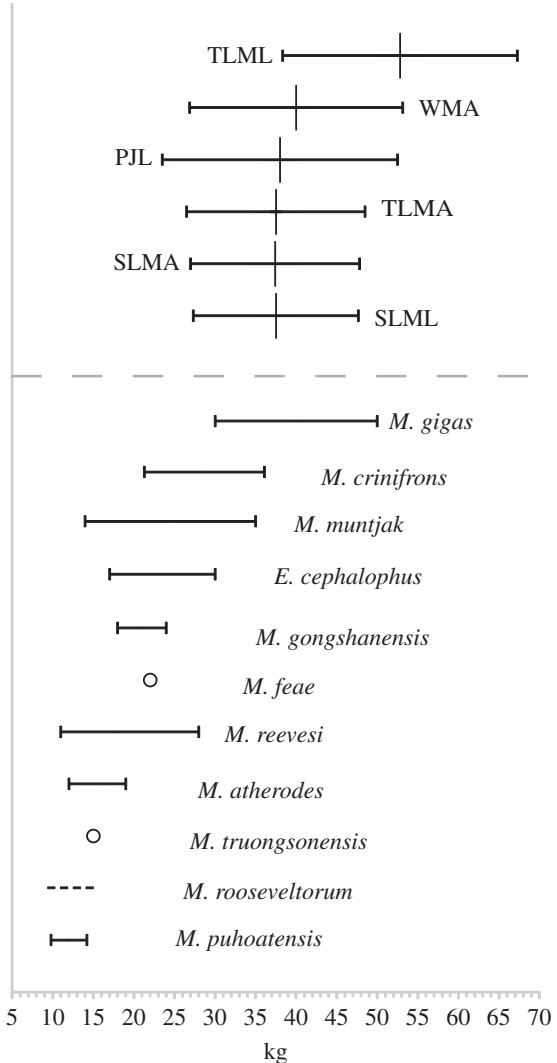

**Figure 6.** Body weight estimates (kg) and %SEE ranges derived from six measurements of specimen HBC-27587 following Janis [42], compared with body weight ranges (bars), single data points (open circles) and estimates (dashed line) for 11 species of the Muntiacini from published sources [23,27,32,56].

between 37 and 53 kg ($n = 6$; mean estimate = 40.5 kg) with percentage standard errors of estimate (%SEE) between 27 and 67 kg. Of the six measurements, five (TLMA, SLMA, SLML, WMA and PJL) produced estimates between 37 and 40 kg (figure 6).

Body weight estimates, absolute size of the specimen and dental metrics indicate against the much smaller *Hydropotes inermis* (water deer: average body weight approx. 14–15 kg, [57]), which (somewhat variably) appear to have a more complex molar morphology with anterior cingulids on the lower molars (figure 3). These metric data also indicate against the larger regional cervids, *Rusa unicolor*, *Cervus nippon*, *Elaphurus davidianus* and *Panolia eldii* (see also [34,50]).

Tooth measurements (figure 5) and body size estimates from HBC-27587 fall well within the lower range of equivalent data for hog deer, *Hyelaphus* (*Axis*) *porcinus* (30–50 kg). A relatively large species/ subspecies *Hyelaphus* (*porcinus*) *annamaticus* is known from the wider region though is now likely to be extirpated in Vietnam [58]. Hog deer molars appear to be relatively high crowned, however, in comparison to the archaeological specimen and the occlusal morphology in *Hyelaphus* appears less 'compact' in the anterior–posterior direction. Critically, occlusal morphology is more complex in *Hyelaphus*. Well-developed anterior cingulids are present on the lower molars (see also [50]). These anterior folds are observable on the occlusal surface and on the labial side of the molars, near to the base of the crown in teeth in well-worn comparative specimens (figure 3). Furthermore, *Hyelaphus* comparatives differ from the archaeological specimen by the presence of a well-developed posterior ectostylid in addition to the anterior ectostylid on the m3 (figure 3).

While extant populations of Siberian roe deer, *Capreolus pygargus*, have a largely temperate distribution and currently range only as far south as Eastern Tibet and central Chinese provinces [59], these are mid-sized cervids (32–48 kg [60]) with similar tooth dimensions and simple occlusal morphology. Average measurements of m2 lengths and widths (SLML, SLMW) from Late Pleistocene samples of '*C. manchuricus*', regarded to be synonymous with *C. pygargus*, from Jilin Province, Northeast China [51] and reference data (NHMUK 83.324) suggest similar dimensions to HBC-27587 (figure 5). Lengths and widths of the third molar are smaller than the archaeological specimen and fall into the upper size range of equivalent data for *M. muntjak* (figure 5). The m2 and m3 lengths of the archaeological specimen exceed those of small samples of Holocene specimens from Primorye [43] but these differences are not significant (SLML: $n = 4$; $t = 1.117$; $p = 0.25$, one-sided; TLML: $n = 5$; $t = 1.291$; $p = 0.09$, one-sided).

Morphologically, the concavity of the posterior margin of the mandibular ramus appears shallower but much wider (i.e extends much further dorsally to the condyle) in *Capreolus* than is suggested in the archaeological specimen. *C. pygargus* lacks anterior cingulids, as is the case with the archaeological specimen, but protoconids and hypoconids are much more angular and there is marked overlap of the metaconids and entoconids. Ectostylids are present on the lower molars, though are weakly developed, particularly on the m3, in comparison to HBC-27587 (figure 3).

The genus *Capreolus* is associated with temperate climates at higher latitudes both in terms of extant populations and the wider fossil record in Eurasia [61]. In Pleistocene records of East Asia, *C. pygargus* does not appear as a component of the *Ailuropoda-Stegodon* fauna [62] and Holocene records (archaeological and historical, $n = 51$) are restricted to central and northern provinces in China, with no records of the species further south than Hubei [11]. As such, *C. pygargus* can also be reasonably discounted on biostratigraphic as well as ecological grounds (see §4.2).

The morphology of the remnant mandible and the simple occlusal morphology of the preserved teeth are consistent with comparative materials from the Muntiacini, a tribe that contains two closely-related genera: *Elaphodus* (tufted deer) and *Muntiacus* (muntjacs). The dental morphology of the lower molars is, in practical terms, very similar if not identical in these taxa [56,63]. The absolute size, dental metrics and body weight estimates from the archaeological specimen, however, indicate that the specimen derived from a large animal for the tribe (figure 4).

The tufted deer, *Elaphodus cephalophus*, is currently restricted to southern China (with historical records from eastern Myanmar). These deer are associated with montane forest habitats and apparently do not range into sub-tropical environments [56]. Hooijer [63] describes a larger, Pleistocene subspecies, *E. cephalophus megalodon*, from Yanjinggou in China. Metrics of individual teeth are not reported, but an upper M1 to M3 length of 410 mm suggest tooth dimensions may be similar to, if not exceeding those of larger *Muntiacus* spp. (cf. [27]). This taxon, however, is known only from early Middle Pleistocene sites in China [64]. Conversely, reported body weight ranges (17–30 kg; figure 6) and metrics of the m2 and m3 (figure 5) indicate that extant *Elaphodus* are relatively small in comparison to equivalent data from the archaeological specimen. Comparison of m3 lengths and widths (TLML, TLMW) with a sample of extant *E. cephalophus* ($n = 9$; this study, [25]) indicate that these dimensions are significantly larger in the archaeological specimen ($T^2 = 26.498$; $p < 0.001$; one-sided).

Within *Muntiacus*, occlusal morphology is relatively simple, as in the archaeological specimen. Anterior cingula, if present on the lower molars, tend to be weakly developed and lost through wear (a single exception, however, was observed on a specimen assigned to *M. muntjak peninsulae*, NHMUK 55.3.2.55, which had a remarkably well-developed anterior cingulid on the m3). Anterior cingula were observed more frequently on *M. muntiacus* than in specimens of *M. reevesi*, even in specimens showing light wear.

The examined *M. muntjak* specimens (from mainland Southeast Asia and variously designated as *M. m. curvostylis*, *grandicornis*, *vaginalis* and *annamensis*) had proportionally weak ectostylids (if present) and a reduced hypoconulid in comparison to the archaeological specimen. Anterior cingula were not observed in *M. crinifrons* and ectostylids, if present, were robust. Anterior cingula were variably present in specimens from *M. feae*: ectostylids, if present, were robust.

A mandible from a young male (NHMUK 2010.484) designated as (*Mega*)*Muntiacus vuquangensis* with the m3 entering occlusion, lacks anterior cingula on the lower molars with weakly developed ectostylids compared to the archaeological specimen. The corresponding skull for this specimen shows relatively long, gracile pedicles rather than the characteristic short, thick pedicles of *M. vuquangensis* and may warrant review. Given that this is a young animal, however, pedicles of the Cervidae shorten and thicken with age [65]. The characteristics of HBC-27587 were a much closer

**Table 3.** Multivariate (modified Hotelling's $T^2$ test) and univariate (modified $t$-test) normative comparisons [46,48] of dental metrics of *Muntiacus* spp. and HBC-27587 with sample datasets of *M. muntjak* from mainland Southeast Asia. Significant differences are shown in italics.

| multivariate normative comparisons: SLML, SLMW, TLML, TLMW. normative sample: *M. muntjak* n = 23 (this study). | | | |
| --- | --- | --- | --- |
| specimen | hypothesis | $T^2$ | p |
| *M. vuquangensis* (NHMUK 210.484) | one-sided (larger) | 1.211 | 0.169 |
| *M. vuquangensis* (AMNH M-274588) | one-sided (larger) | 2.269 | *0.05* |
| HBC-27587 (this study) | one-sided (larger) | 2.468 | *0.04* |
| *M. feae* (AMNH 32.11.1.17.1) | two-sided (difference) | 0.642 | 0.639 |
| *M. feae* (AMNH 24.1.6.2) | two-sided (difference) | 1.888 | 0.154 |
| *M. crinifrons* (AMNH M-56991) | two-sided (difference) | 0.859 | 0.506 |
| *M. crinifrons* (NHMUK 1.3.2.21) | two-sided (difference) | 0.985 | 0.439 |
| **Multivariate normative comparisons: TLML, TLMW.** **Normative sample: *M. muntjak* n = 32 (this study, [26]).** | | | |
| specimen | hypothesis | $T^2$ | p |
| HBC-27587 (this study) | one-sided (larger) | 4.966 | *0.007* |
| **Univariate normative comparisons: SLML and TLML.** **normative samples: SLML *M. muntjak* n = 23 (this study) and TLML *M. muntjak* n = 32 (this study, [26]).** | | | |
| specimen | hypothesis | t | p |
| SLML *M. gigas* (T301-6) | one-sided (larger) | 0.497 | 0.318 |
| TLML *M. gigas* (T301-6) | one-sided (larger) | 0.266 | 0.394 |

morphological match with a mandible of *M. vuquangensis* from a 34 kg female [27] collected by George Schaller in Lao PDR (AMNH M-274588). The concave posterior margin of the remnant mandible was consistent with the comparative specimen and the lower molars of the Lao PDR specimen were worn, are proportionally robust and lack anterior cingula. The hypoconulid of the m3 is well-developed and rounded. Well-developed and robust ectostylids are present on the m2 and m3, as in the archaeological specimen (figure 3). The morphological characteristics of HBC-27587 are consistent with Muntiacini, with a close match to available comparative material from *M. vuquangensis*.

Length and width measurements of the m2 (SLML, SLMW) and m3 (TLML, TLMW) from available comparatives for the Muntiacini form two major clusters with a marginal overlap: the first, and smallest in terms of dimensions, are measurements from *M. reevesi* and secondly, measurements from subspecies of *M. muntjak* from Mainland Southeast Asia. Measurements from extant *E. cephalophus* fall in the lower size range of the *M. muntjak* clusters (figure 5).

The limited available measurements from larger-bodied *Muntiacus* spp., *M. crinifrons* (n = 2) and *M. feae* (n = 2) fall within the upper range of *M. muntjak*. Multivariate normative comparisons of m2 and m3 lengths and widths (SLML, SLMW, TLML, TLMW) of available data from these species (n = 2, in both cases) with sample data from specimens of *M. muntjak* from mainland Southeast Asia (n = 23) indicate no significant difference (table 3).

Available data from two specimens of '*M. vuquangensis*', the young male (NHMUK 2010.484) and the 34 kg female (AMNH M-274588), indicate that measurements from giant muntjac overlap with the upper range of *M. muntjak* (figure 5). Lengths of the m2 and m3 (SLML = 12.58 mm and TLML = 16.71 mm, respectively) from a specimen of *M. gigas* (T301-6) from Tianluoshan, near Hemudu (Turvey, unpublished data) also fall within the range of *M. muntjak*. Multivariate normative comparisons of m2 and m3 lengths and widths (SLML, SLMW, TLML, TLMW) of the young male *M. vuquangensis* with sample data from *M. muntjak* (n = 23) indicated no significant difference (table 3). This test indicated that the metrics from Schaller's female specimen are marginally, but significantly, larger than the normative sample (table 3). Comparison of m2 length (SLML) and m3 length (TLML) from the *M. gigas* specimen indicated no significant difference between these measures and the normative sample of *M. muntjak*.

The dimensions of the m2 and m3 of HBC-27587 exceed all available comparative data from species within the tribe (figure 5). Comparisons of m2 and m3 lengths and widths ($n = 23$) and m3 lengths and widths ($n = 32$) of HBC-27587 indicate that these variables are significantly larger in the archaeological specimen, than in the sample data for *M. muntjak* (table 3). The body weight estimates derived from measurements of the mandible are also consistent with the trend in dental metrics and suggest a larger animal than the majority of *Muntiacus* spp. A mean estimate of approximately 40 kg for the specimen exceeds reported weight ranges of all taxa within the tribe bar one: *M. gigas* (figure 6). While the dental dimensions of the Hang Boi specimen exceed available comparative data, given the range of variation within dental measurements in the genus (figure 5) there is no reason to suggest that the archaeological specimen would be an excessively or anomalously large specimen for this taxon. As such, there are no compelling grounds to posit a novel species and we refer specimen HBC-27587 to *M. gigas* (syn. *vuquangensis*).

# 4. Discussion

## 4.1. Quaternary records of *Muntiacus*

Quaternary records of *Muntiacus* spp. from archaeological and palaeontological sites in East and Southeast Asia are not uncommon (e.g. [25,26,66–68]; figure 1) but explicit references to giant muntjacs are rare. Analyses of mitochondrial DNA suggest that giant muntjacs appeared in the Early Pleistocene between 0.9 and 1.8 million years ago [29]. At present, however, the only confirmed giant muntjac fossils derive from one Late Pleistocene site (Yuhang; figure 1; site 8) and a handful of Holocene archaeological sites in China, collectively dating between 2.2 and 7 ka [7]. These specimens indicate a much wider post-glacial range for populations of giant muntjac than the records from the Annamite range suggest (figure 1; sites 1–8). The new record from Hang Boi (figure 1; site 12) is consistent with this observation. It is the first subfossil record from Vietnam and it extends the current known Holocene geographical range for the species in the country. Hang Boi is located more than 150 km north of the current reported range of *M. vuquangensis* (figure 1) although, in the context of the Chinese records, this is not a dramatic expansion of the Holocene range of the species.

But what of potential records of giant muntjac in deeper time? Dental remains attributed to *Muntiacus* spp. are a common component in archaeological and palaeontological investigations in the region, but the possibility of collections containing dental remains of giant muntjac is rarely considered [7]. The relationship between dental dimensions and body size should not be treated uncritically but there is a well-defined and logical trend between these factors [42,69,70] and a defining characteristic of giant muntjac is large stature. Published dental data from earlier, regional Pleistocene records of teeth attributed to *Muntiacus* indicate that HBC-27587 is not unique (figure 7). This raises the possibility that further as-yet unidentified giant muntjac specimens are present in Quaternary archives from the region, which could provide further spatial and temporal information for the species. This is likely to be a complex issue to address at present and a matter of interpretation.

Firstly, given the uncertainties surrounding the taxonomy of *Muntiacus* [24] and apparently rapid, recent diversification within the genus (for example, analyses of mitochondrial DNA indicate that *M. muntjak* and *M. feae* appeared in the Middle Pleistocene [29]) the identification and reporting of *Muntiacus*, in many cases individual teeth, is complex. Given the size overlap indicated by comparative data from *M. muntjak*, *M. crinifrons*, *M. feae* and specimens of *M. gigas* (*vuquangensis*), (figure 5), in many cases metric identifications of individual teeth may only realistically be attributed to *Muntiacus* sp.: the use of '*M. muntjak*' as a catch-all taxon should probably be abandoned in the archaeological and palaeontological literature.

Secondly, published metric identifications from Middle and Late Pleistocene sites in the region have been interpreted to suggest Pleistocene populations of *Muntiacus* were larger-bodied than post-glacial populations [26,63]. While there can be difficulties with identifying individual cervid teeth [25] and the potential of inter-observer variance in measurements need to be taken into account [71], data from Southeast (figure 1: sites 11, 13–15) and East Asia (figure 1: sites 9–10) appear compelling [25,26]. A comparison of Middle and Late Pleistocene m3 lengths from specimens attributed to '*M. muntjak*' with data from extant *M. muntjak* ($n = 32$; this study, [26]) does suggest larger size in the Pleistocene (figure 7): the difference in median values is statistically significant (Kruskal–Wallace: $H_c$ (tie-corrected) $= 43.03$; $p < 0.001$). Conversely, comparison of median values of Middle and Late

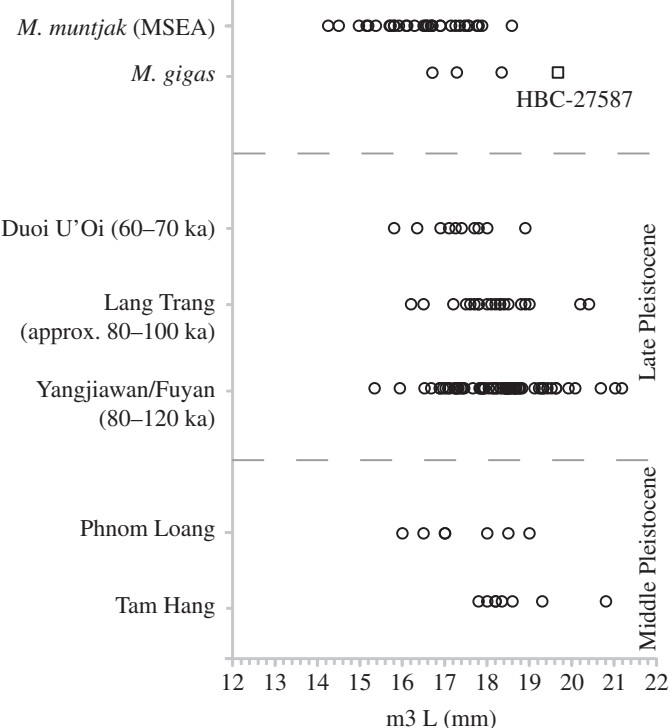

**Figure 7.** Length ranges of lower third molars (m3 L) for records of *Muntiacus* from Middle and Late Pleistocene regional sites [25,26] compared with comparative data from extant *M. muntjak* (Mainland Southeast Asia), *M. gigas* and HBC-27587 (this study).

Pleistocene records and *M. gigas* ($n = 4$; including HBC-27587) indicated no significant difference (Kruskal–Wallace: $H_c$ (tie-corrected) $= 0.2468$; $p = 0.8839$).

In the Pleistocene datasets, m3 length does not appear to vary as a function of time between Middle and Late Pleistocene (Mann Whitney $U = 702$; $p = 0.7373$), although sample sizes are uneven (Late Pleistocene: $n = 99$; Middle Pleistocene: $n = 15$). Median values also do not appear to differ as a function of latitude (Late Pleistocene Vietnam, $n = 30$; Late Pleistocene China, $n = 69$; Mann Whitney $U = 786$; $p = 0.0597$). While there is an internal consistency within the Pleistocene records, critically, the potential impact of the presence of unrecognized dental remains of the largest known species of muntjac in these datasets has not been considered. As such, a larger comparative dataset from *M. gigas*, which characterizes the metric characteristics of this taxon more fully, would be an invaluable step not only in the investigation of the Quaternary history of the species but also in the assessment of Pleistocene trends in the genus.

## 4.2. Implications for extant populations

At present, the Tràng An karst is vegetated by highly-adapted edaphic limestone forest, which incorporates evergreen and semi-evergreen components. Given that the mandible derives from a deposit dated to 11.3 ka, consideration of the early Holocene palaeoenvironmental context of the specimen may be instructive for understanding habitat requirements of the giant muntjac.

Current data on the habitat preferences of the species are limited, but giant muntjac are not known to occur in the drier semi-evergreen hill forests found to the north and west of the Annamite region. The species has been recorded in mosaic habitats, but available information suggests a preference for closed-canopy forest, predominantly the evergreen and semi-evergreen forests characteristic of the Annamites, including areas with steep terrain. The species has been recorded up to 1200 m above sea level (asl) but available data suggests the species is most likely to occur below 1000 m asl. Reliable estimates of lower elevational limits, however, are confounded by recent trends in habitat removal and over-hunting [33]. The Tràng An karst rarely approaches 200 m asl and the coastal plain that surrounds the massif, and dolines within it, are only a few metres above sea level. The specimen from Hang Boi indicates that giant muntjac occurred in a lowland karst setting and, as such, the current altitudinal range of the giant muntjac in the Annamites may not represent an ecological preference for habitats at higher altitudes, but rather a response to human activities.

Early Holocene proxy palaeoenvironmental records from terrestrial sites within the region are relatively sparse but it is clear that these records track global trends. For example, palaeoclimatic proxies from stalagmites [72] and lake cores [73–75] in southern China show strong consilience with records from Greenland [76] and the Caribbean [77]. Well-documented climatic events during the most recent deglaciation, such as the Bølling-Allerød interstadial (increased temperature/precipitation approx. 14.7–12.9 ka) and Younger Dryas stadial (reduced temperature/precipitation approx. 12.9–11.7 ka) are visible.

$\delta^{18}$O speleothem records from Tham Duon Mai Cave in northern Laos [78] show marked similarity with those of Chinese speleothems and record variations in East and Southeast Asian rainfall, driven by summer monsoon strength. In contrast to the Chinese records, however, the $\delta^{13}$C record from Tham Duon Mai Cave is indicative of a drier climate at the beginning of the Holocene in mainland Southeast Asia. This contrast can also be seen when comparing the pollen record from Huguangyan Maar Lake in coastal southern China [79] with the pollen sequence from Nong Pa Kho in northeast Thailand [80]. At Huguangyan Maar Lake, the Younger Dryas is represented by decreasing concentrations of evergreen taxa in general, but increases in the percentages of cold-tolerant herbs. The evergreen genus *Quercus* maintains dominance, however, suggestive of a cool/wet climatic regime. Increases in temperature and precipitation across the ensuing early Holocene, coincident with the insolation maximum, are borne out by the expansion of tropical evergreen broad-leaved forest [79]. In northeast Thailand, pollen zones 5 and 6 from Nong Pa Kho [80] correspond to the period from approximately 14.7–8.9 ka and represent the end of a relatively long period at the end of the Pleistocene during which a mosaic of dry deciduous taxa, humid forest elements and more temperate components persisted. Evidence from pollen and other botanical remains recovered from the nearby site (approx. 30 km) of Con Moong Cave [81] also suggest that the closest analogue to local habitats at that time are likely to have been dry, open woodland mosaic with components of moister forest types.

Within Tràng An, pollen sequences and lipid biomarkers recovered from a nearby (approx. 1 km) cave site, Hang Trong, date between 37 and 14 ka [82] and therefore predate the sequence at Hang Boi. The record from Hang Trong is instructive, however, as this time period incorporates the Last Glacial Maximum (LGM; 26–18 ka). Regionally, the LGM was characterized by lower temperatures, increased aridity and lowered sea levels with the exposure of the Sunda shelf [83]. Despite regional aridity and lowered temperatures the pollen and lipid biomarker sequences from Hang Trong indicate the presence of limestone karst forest taxa throughout this period [82].

The animal taxa that have been identified from bones in the same stratigraphic deposit as the mandible from Hang Boi also represent a line of proxy palaeoenvironmental evidence. The bones, which are the refuse from human subsistence activities, suggest that human activity focused on the exploitation of forested habitats. Identified specimens include several (cut-marked) fragments of post-cranial elements from hog badger (*Arctonyx collaris*), a maxilla fragment from a large male macaque (*Macaca* sp.—most likely attributable to *M. assamensis* or *M. arctoides*), post-cranial remains of relatively large colobine monkeys, a large bovid (*Bos* cf. *gaurus*), sambar (*Rusa unicolor*) and civet (*Paradoxurus hermaphroditus*).

In summary, current palaeoenvironmental evidence suggests a drier regional climate at the onset of the Holocene and that the landscapes around Tràng An likely comprised drier, open woodland habitats although limestone forest persisted on the karst. From depositional context, HBC-27587 is most likely the remains of an animal that was consumed. Thus, a whole or partial carcass was initially transported to the site. It is unlikely, however, that the animal was transported more than 10 km from Hang Boi. Meat is perishable and forested limestone karst is a difficult and time-consuming habitat to navigate. Whether the animal was secured (either hunted or scavenged) from within the karst forest or in one of the many dolines in the local landscape is a matter of speculation. Given the contextual relationship with animal taxa that have associations with forest habitats, however, it is most likely that the Hang Boi muntjac was derived locally to the cave site, within the karst forest. As a matter of speculation, Tràng An may have represented a refugium for forest dependant taxa, including giant muntjac, in the face of a drier climate and more open woodland habitats in the region at the beginning of the Holocene.

# 5. Conclusion

The subfossil mandible of *Muntiacus gigas* (syn. *vuquangensis*) from Hang Boi (HBC-27587), dated to 11 300 years before present, extends the known spatial and temporal range of giant muntjacs in Vietnam. The specimen is further evidence that giant muntjacs were much more widely distributed in Holocene East Asia than historical records have indicated. Our find lends further support to the

assertion that elements of the novel fauna of the Annamite region are likely to represent refugial populations that have been (and continue to be) subject to anthropogenic pressures, rather than denoting a centre of endemism. This, however, in no way detracts from the urgent need to conserve remaining populations and habitats, which are currently in rapid decline.

By the time the giant muntjac was described scientifically (both as subfossil and in life) it was highly likely that populations were already much reduced over its former range. Indeed, while the new specimen suggests that giant muntjacs have been exploited for over 11 000 years in Vietnam, it is important to reiterate the differences between human demography today and a hunting and gathering technological base, over 11 000 years ago, and the decline that the species has experienced since scientific description. The current pressures on the giant muntjac of large-scale, rapid habitat modification and removal, rapid increase in human population and the extraction of animal fauna from remnant forests are novel and recent occurrences in the known human history of the region and are not sustainable.

Data accessibility. The datasets supporting this article have been uploaded as electronic supplementary material.
Authors' contributions. C.M.S. and R.J.R conceived the study. C.M.S. conducted analyses and identification and drafted the manuscript with contributions from all authors; C.M.S. and B.U. collected comparative data; R.J.R., C.M.S., T.K., B.V.M. and P.S.K. carried out excavation, site analysis and recording; S.O. and N.T.M.H. produced the palaeoenvironmental synthesis. All authors revised the manuscript and gave final approval for publication.
Competing interests. We declare we have no competing interests.
Funding. Funding for this project was provided by the Arts and Humanities Research Council (Global Challenges Research Fund) award: AH/N005902/1 and by the Xuan Truong Enterprise.
Acknowledgements. We thank the Tràng An Management Board, the Ninh Binh Provincial People's Committee and Xuan Truong Enterprise for ongoing support for research in the Tràng An World Heritage Area. The study would not have been possible without access to comparative museum collections at the Oxford University Museum of Natural History (Darren Mann, Mark Carnall, Eileen Westwig and Chris Jarvis), University Museum of Zoology, Cambridge (Matthew Lowe), American Museum of Natural History (Eleanor Hoeger and Marisa Surovy) and Natural History Museum, UK (Roberto Portela Miguez). We also thank Minghao Lin for his assistance in locating sources and three anonymous reviewers for their comments and suggestions, which improved the content of the manuscript.

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
