## [Reviewer comments · Royal Society Open Science]

Review History

RSOS-181461.R0 (Original submission)

Review form: Reviewer 1

Is the manuscript scientifically sound in its present form?

No

Are the interpretations and conclusions justified by the results?

Yes

Is the language acceptable?

Yes

Is it clear how to access all supporting data?

Not Applicable

Do you have any ethical concerns with this paper?

No

Have you any concerns about statistical analyses in this paper?

Yes

Recommendation?

Major revision is needed (please make suggestions in comments)

Comments to the Author(s)

The paper by Stimpson et al. provides important new historical baseline data on a Critically Endangered but largely overlooked mammal species, with potentially important implications for understanding its ecological requirements and thus informing future conservation management. However, additional analysis and some restructuring of the manuscript is required before it can be published.

Major comments:

- The introduction is all a bit too specific, and makes the paper feel like it would only appeal to someone already interested in muntjacs. It would be much better to open the paper by framing the study in a much wider context, by starting off with an explanation about the importance of using data from the past (historical baselines, environmental archives) to provide new insights for conservation of threatened species, and the ways in which this can be / has been done - i.e. make the paper of interest to people beyond just muntjac specialists!
- Similarly, the previous suggestion that vuquangensis and gigas are the same species should be introduced in the introduction, rather than this important point being relegated to the discussion - as your study provides further evidence of a past wider distribution of giant muntjacs, which supports the previous study.
- Similarly, the idea that the name "gigas" can be "ignored" for now, even though it represents a senior synonym of vuquangensis, because "this revision has not yet been formally adopted", is a bit of a red herring - the way it can be adopted is by studies like this deciding to use the name "gigas" (it doesn't require a formal IUCN edict) - as there is no worldwide arbiter of taxonomy. So, if the authors consider that vuquangensis and gigas are the same taxon, they should use the name gigas in this manuscript; and if not, then they should state why they think they're different.
- When comparing measurements between the new specimen and existing data for other muntjacs, the authors should also compare their specimen with available measurement data for Muntiacus gigas - this is really important to do.
- The lack of any statistical analysis to compare the new specimen with other muntjac samples (muntiacus, vuquangensis, gigas) is a major omission, and really has to be done - even if just via something fairly straightforward such as t-tests or ANOVAs. A statistical assessment of measurement differences, rather than just "eyeballing" the data, is essential.

Minor comments:

- "sub-fossil" should be changed throughout the manuscript to "subfossil" (the more normal way of spelling this word)
- "critically endangered" should be capitalised to "Critically Endangered", as this is a formal IUCN threat category. It also shouldn't be written in inverted commas (line 58)
- Line 112: "11, 778" - spacing after the comma needs correcting

- Line 243: "ectosylids" spelled incorrectly
- Line 257: "Muntiacini" shouldn't be italicised
- Line 263: "celaphodus" spelled incorrectly
- Line 285: not strictly true that the only confirmed gigas specimens derive from Holocene sites in China - Turvey et al 2016 also included a Late Pleistocene specimen from Yuhang in their analyses, so Pleistocene material has been interpreted by previous authors as representing gigas
- Line 352-3: even if lower elevational estimates are confounded by human activity, it would still be useful to say what the current lower elevational range estimate is for the species, and how much this differs from the elevation of the Holocene site from which the new specimen was found - and also discuss the idea of the Annamites acting as an altitudinal refugium for the species, in the context of this concept more widely for SE Asian and other mammal species

Review form: Reviewer 2 (Gertrud Rössner)

Is the manuscript scientifically sound in its present form?

Yes

Are the interpretations and conclusions justified by the results?

Yes

Is the language acceptable?

Yes

Is it clear how to access all supporting data?

Yes

Do you have any ethical concerns with this paper?

No

Have you any concerns about statistical analyses in this paper?

No

Recommendation?

Accept with minor revision (please list in comments)

Comments to the Author(s)

Dear authors,

This is a carefully conducted transdisciplinary study, a very well written manuscript, and an excellent example that we need the big picture, far beyond historical time, when assessing conservational needs. Congratulations, I've enjoyed reading very much!

I've only indicated minor suggestions to consider in the manuscript.

The two most important among them, I'm coming up with here again:

Critical data of *Capreolus pygargus* and *Elaphodus cephalophus megalodon* seem to have been

neglected. In order to make the issues more transparent I'd suggest to add relevant data.
Two issue

Review form: Reviewer 3

Is the manuscript scientifically sound in its present form?

No

Are the interpretations and conclusions justified by the results?

Yes

Is the language acceptable?

Yes

Is it clear how to access all supporting data?

No

Do you have any ethical concerns with this paper?

No

Have you any concerns about statistical analyses in this paper?

No

Recommendation?

Accept with minor revision (please list in comments)

Comments to the Author(s)

The manuscript is well written and if the topic falls into the scope of the journal, it definitely is worth to be published. Unfortunately I have not found any access to the supplementary tables and figures. Therefore, I could not comment some of the evidence used in the ms. Substantial part of the arguments is based on the body mass estimation. The method is recalled from Janis (1990). Since the reference deals with a chapter from a book published almost 30 years ago, it may be difficult for many readers to get and read the methodology there. This may have two implications to the recent manuscript. First, the authors of the present study should describe in brief the principles of the method for those readers who do not have the cited book at hand. Second, from a more recent literature (e. g., Mendoza, M., Janis, C. M. & Palmqvist, P., 2006. Estimating the body mass of extinct ungulates: a study on the use of multiple regression. *J. Zool.* 270, 90-101; De Esteban-Trivigno, S. & Köhler, M., 2011. New equations for body mass estimation in bovids: Testing some procedures when constructing regression functions. *Mamm. Biol.* 76, 755-761) it is apparent that some improvements of the method are available since 1990. Even if it was not applicable to the paleo-material available, it seems the body mass estimation based on multiple regression should be considered more carefully.

Ad Table 1 - These are important measures. A sketch showing simply how the mandible was measured would help.

Ad Figure 3 - Chinese water deer announced in the text (lines 196-198) is absent in the figure.

Decision letter (RSOS-181461.R0)

17-Dec-2018

Dear Dr Stimpson,

The editors assigned to your paper ("An 11,000-year-old giant muntjac (*Muntiacus vuquangensis*) sub-fossil from Northern Vietnam: implications for past and present populations") have now received comments from reviewers. We would like you to revise your paper in accordance with the referee and Associate Editor suggestions which can be found below (not including confidential reports to the Editor). Please note this decision does not guarantee eventual acceptance.

Please submit a copy of your revised paper before 09-Jan-2019. Please note that the revision deadline will expire at 00.00am on this date. If we do not hear from you within this time then it will be assumed that the paper has been withdrawn. In exceptional circumstances, extensions may be possible if agreed with the Editorial Office in advance. We do not allow multiple rounds of revision so we urge you to make every effort to fully address all of the comments at this stage. If deemed necessary by the Editors, your manuscript will be sent back to one or more of the original reviewers for assessment. If the original reviewers are not available, we may invite new reviewers.

- Data accessibility

If you wish to submit your supporting data or code to Dryad (<http://datadryad.org/>), or modify your current submission to dryad, please use the following link:
<http://datadryad.org/submit?journalID=RSOS&manu=RSOS-181461>

- Competing interests

- Authors' contributions

- Acknowledgements

- Funding statement

Kind regards,

Andrew Dunn

on behalf of Prof Kevin Padian (Subject Editor)

Comments to Author:

Reviewers' Comments to Author:

Reviewer: 1

Comments to the Author(s)

The paper by Stimpson et al. provides important new historical baseline data on a Critically Endangered but largely overlooked mammal species, with potentially important implications for

understanding its ecological requirements and thus informing future conservation management. However, additional analysis and some restructuring of the manuscript is required before it can be published.

Major comments:

- The introduction is all a bit too specific, and makes the paper feel like it would only appeal to someone already interested in muntjacs. It would be much better to open the paper by framing the study in a much wider context, by starting off with an explanation about the importance of using data from the past (historical baselines, environmental archives) to provide new insights for conservation of threatened species, and the ways in which this can be / has been done - i.e. make the paper of interest to people beyond just muntjac specialists!

- Similarly, the previous suggestion that *vuquangensis* and *gigas* are the same species should be introduced in the introduction, rather than this important point being relegated to the discussion - as your study provides further evidence of a past wider distribution of giant muntjacs, which supports the previous study.

- Similarly, the idea that the name "*gigas*" can be "ignored" for now, even though it represents a senior synonym of *vuquangensis*, because "this revision has not yet been formally adopted", is a bit of a red herring - the way it can be adopted is by studies like this deciding to use the name "*gigas*" (it doesn't require a formal IUCN edict) - as there is no worldwide arbiter of taxonomy. So, if the authors consider that *vuquangensis* and *gigas* are the same taxon, they should use the name *gigas* in this manuscript; and if not, then they should state why they think they're different.

- When comparing measurements between the new specimen and existing data for other muntjacs, the authors should also compare their specimen with available measurement data for *Muntiacus gigas* - this is really important to do.

- The lack of any statistical analysis to compare the new specimen with other muntjac samples (*muntiacus*, *vuquangensis*, *gigas*) is a major omission, and really has to be done - even if just via something fairly straightforward such as t-tests or ANOVAs. A statistical assessment of measurement differences, rather than just "eyeballing" the data, is essential.

Minor comments:

- "sub-fossil" should be changed throughout the manuscript to "subfossil" (the more normal way of spelling this word)

- "critically endangered" should be capitalised to "Critically Endangered", as this is a formal IUCN threat category. It also shouldn't be written in inverted commas (line 58)

- Line 112: "11, 778" - spacing after the comma needs correcting

- Line 243: "ectosylids" spelled incorrectly

- Line 257: "Muntiacini" shouldn't be italicised

- Line 263: "celaphodus" spelled incorrectly

- Line 285: not strictly true that the only confirmed *gigas* specimens derive from Holocene sites in China - Turvey et al 2016 also included a Late Pleistocene specimen from Yuhang in their analyses, so Pleistocene material has been interpreted by previous authors as representing *gigas*

- Line 352-3: even if lower elevational estimates are confounded by human activity, it would still be useful to say what the current lower elevational range estimate is for the species, and how much this differs from the elevation of the Holocene site from which the new specimen was found - and also discuss the idea of the Annamites acting as an altitudinal refugium for the species, in the context of this concept more widely for SE Asian and other mammal species

Reviewer: 2

Comments to the Author(s)

Dear authors,

This is a carefully conducted transdisciplinary study, a very well written manuscript, and an excellent example that we need the big picture, far beyond historical time, when assessing conservational needs. Congratulations, I've enjoyed reading very much!

I've only indicated minor suggestions to consider in the manuscript.

The two most important among them, I'm coming up with here again:

Critical data of *Capreolus pygargus* and *Elaphodus cephalophus megalodon* seem to have been neglected. In order to make the issues more transparent I'd suggest to add relevant data.

Two issue

Reviewer: 3

Comments to the Author(s)

The manuscript is well written and if the topic falls into the scope of the journal, it definitely is worth to be published. Unfortunately I have not found any access to the supplementary tables and figures. Therefore, I could not comment some of the evidence used in the ms. Substantial part of the arguments is based on the body mass estimation. The method is recalled from Janis (1990). Since the reference deals with a chapter from a book published almost 30 years ago, it may be difficult for many readers to get and read the methodology there. This may have two implications to the recent manuscript. First, the authors of the present study should describe in brief the principles of the method for those readers who do not have the cited book at hand. Second, from a more recent literature (e. g., Mendoza, M., Janis, C. M. & Palmqvist, P., 2006. Estimating the body mass of extinct ungulates: a study on the use of multiple regression. *J. Zool.* 270, 90-101; De Esteban-Trivigno, S. & Köhler, M., 2011. New equations for body mass estimation in bovids: Testing some procedures when constructing regression functions. *Mamm. Biol.* 76, 755-761) it is apparent that some improvements of the method are available since 1990. Even if it was not applicable to the paleo-material available, it seems the body mass estimation based on multiple regression should be considered more carefully.

Ad Table 1 - These are important measures. A sketch showing simply how the mandible was measured would help.

Ad Figure 3 - Chinese water deer announced in the text (lines 196-198) is absent in the figure.

Author's Response to Decision Letter for (RSOS-181461.R0)

See Appendix A.

RSOS-181461.R1 (Revision)

Review form: Reviewer 1

Is the manuscript scientifically sound in its present form?

Yes

Are the interpretations and conclusions justified by the results?

Yes

Is the language acceptable?

Yes

Is it clear how to access all supporting data?

Yes

Do you have any ethical concerns with this paper?

No

Have you any concerns about statistical analyses in this paper?

No

Recommendation?

Accept with minor revision (please list in comments)

Comments to the Author(s)

The authors have done a great job in addressing all of my comments and suggestions, and this is a great paper - I look forward to it being published! The only very minor further edits that I'd suggest are a few typos here and there:

- Page 3, lines 56-58: It feels like there's a word missing from this sentence - maybe it needs a "For" at the start of the sentence?
- Page 5, line 47: should "Primoye" be "Primorye"?
- Page 5, line 57: "temperature" should be "temperate"
- Page 6, line 4: remove "and" after "biostratigraphic"
- Page 6, line 50: should "(NHMUK 2010)" be removed? (not sure what this means after the actual reported specimen number)
- Page 7, line 49: remove "of" from before "are present in Quaternary archives"
- Page 8, line 3: remove "of" after "published metric identifications"
- Page 8, line 6: "post glacial" is usually either hyphenated or one word
- Figure 3: "Hydropotes" is spelled wrong in the figure, and "Capreolus (capreolus) pygargus" should have the subgenus name in parentheses removed (since it's incorrectly capitalised, and you don't refer to this subgenus name in the main text)

Decision letter (RSOS-181461.R1)

12-Feb-2019

Dear Dr Stimpson:

On behalf of the Editors, I am pleased to inform you that your Manuscript RSOS-181461.R1 entitled "An 11,000-year-old giant muntjac subfossil from Northern Vietnam: implications for past and present populations" has been accepted for publication in Royal Society Open Science subject to minor revision in accordance with the referee suggestions. Please find the referees' comments at the end of this email.

The reviewers and Subject Editor have recommended publication, but also suggest some minor revisions to your manuscript. Therefore, I invite you to respond to the comments and revise your manuscript.

- Ethics statement

- Data accessibility

<http://datadryad.org/submit?journalID=RSOS&manu=RSOS-181461.R1>

- Competing interests

- Authors' contributions

- Acknowledgements

- Funding statement

Because the schedule for publication is very tight, it is a condition of publication that you submit the revised version of your manuscript before 21-Feb-2019. Please note that the revision deadline will expire at 00.00am on this date. If you do not think you will be able to meet this date please let me know immediately.

Supplementary files will be published alongside the paper on the journal website and posted on

the online figshare repository (<https://figshare.com>). The heading and legend provided for each supplementary file during the submission process will be used to create the figshare page, so please ensure these are accurate and informative so that your files can be found in searches. Files on figshare will be made available approximately one week before the accompanying article so that the supplementary material can be attributed a unique DOI.

on behalf of Prof Kevin Padian (Subject Editor)
openscience@royalsociety.org

Reviewer comments to Author:
Reviewer: 1

Comments to the Author(s)

The authors have done a great job in addressing all of my comments and suggestions, and this is a great paper - I look forward to it being published! The only very minor further edits that I'd suggest are a few typos here and there:

- Page 3, lines 56-58: It feels like there's a word missing from this sentence - maybe it needs a "For" at the start of the sentence?
- Page 5, line 47: should "Primoye" be "Primorye"?
- Page 5, line 57: "temperature" should be "temperate"
- Page 6, line 4: remove "and" after "biostratigraphic"
- Page 6, line 50: should "(NHMUK 2010)" be removed? (not sure what this means after the actual reported specimen number)
- Page 7, line 49: remove "of" from before "are present in Quaternary archives"
- Page 8, line 3: remove "of" after "published metric identifications"
- Page 8, line 6: "post glacial" is usually either hyphenated or one word
- Figure 3: "Hydropotes" is spelled wrong in the figure, and "Capreolus (capreolus) pygargus" should have the subgenus name in parentheses removed (since it's incorrectly capitalised, and you don't refer to this subgenus name in the main text)

Author's Response to Decision Letter for (RSOS-181461.R1)

See Appendix B.

Decision letter (RSOS-181461.R2)

18-Feb-2019

Dear Dr Stimpson,

I am pleased to inform you that your manuscript entitled "An 11,000-year-old giant muntjac subfossil from Northern Vietnam: implications for past and present populations" is now accepted for publication in Royal Society Open Science.

on behalf of Professor Kevin Padian (Subject Editor)
openscience@royalsociety.org

Follow Royal Society Publishing on Twitter: [@RSocPublishing](https://twitter.com/RSocPublishing)
Follow Royal Society Publishing on Facebook:
<https://www.facebook.com/RoyalSocietyPublishing.FanPage/>
Read Royal Society Publishing's blog: <https://blogs.royalsociety.org/publishing/>

Appendix A

RESPONSE TO REVIEWERS

We thank the referees and editors for their time reviewing our submission “*An 11,000-year-old giant muntjac subfossil from Northern Vietnam: implications for past and present populations*”. The comments and suggested revisions improve the content of the revised manuscript and the acknowledgements have been updated, accordingly. We are particularly grateful to Reviewer 1 for supplying details of the dental dimensions of *Muntiacus gigas*.

1. REVIEWER 1

The paper by Stimpson et al. provides important new historical baseline data on a Critically Endangered but largely overlooked mammal species, with potentially important implications for understanding its ecological requirements and thus informing future conservation management. However, additional analysis and some restructuring of the manuscript is required before it can be published.

1.1 Comment

The introduction is all a bit too specific, and makes the paper feel like it would only appeal to someone already interested in muntjacs. It would be much better to open the paper by framing the study in a much wider context, by starting off with an explanation about the importance of using data from the past (historical baselines, environmental archives) to provide new insights for conservation of threatened species, and the ways in which this can be / has been done - i.e. make the paper of interest to people beyond just muntjac specialists!

Response

We agree with this comment and the study would benefit from being placed in a wider research context. The introduction has been revised and expanded with the addition of the following section:

PAGE 1 lines 44-59

“Human activities continue to reduce mammal populations, geographic ranges and, ultimately, cause extinctions [1]. While the scale, rapidity and mechanisms driving recent anthropogenic impacts such as these are unprecedented [2-3], attempts to form effective conservation strategies are potentially hampered by a paucity of studies that consider biological communities from millennial as well as ecological (typically less than 50 years) timescales [2,4-7]. Current studies and recent data (less than 100 years) characterise animal communities and their habitats only after, potentially, centuries or millennia of exploitation and modification by humans [7-10]. In this sense, a historical amnesia results in a shifting baseline syndrome where ecosystems, animal populations and their current geographic distribution are interpreted as ‘natural’ or pristine, where they are in fact degraded [2,9,11]. This issue is likely to be particularly problematic with rare, recently described and poorly-known mammals [7,11]. In this context, the potential of Quaternary archaeological and palaeontological data to provide longer time-scale perspectives and benchmark evidence for biological conservation is increasingly being recognised and demonstrated [7,9-16]. Here, we present just such a line of evidence and consider a poorly-known and Critically Endangered species of deer (Cervidae): the giant muntjac.”

The bibliography has been updated accordingly, references 1-16.

1.2 Comment

*Similarly, the previous suggestion that *vuquangensis* and *gigas* are the same species should be introduced in the introduction, rather than this important point being relegated to the discussion – as your study provides further evidence of a past wider distribution of giant muntjacs, which supports the previous study.*

*Similarly, the idea that the name “*gigas*” can be “ignored” for now, even though it represents a senior synonym of *vuquangensis*, because “this revision has not yet been formally adopted”, is a bit of a red herring – the way it can be adopted is by studies like this deciding to use the name “*gigas*” (it doesn’t require a formal IUCN edict) – as there is no worldwide arbiter of taxonomy. So, if the authors consider that *vuquangensis* and *gigas* are the same taxon, they should use the name *gigas* in this manuscript; and if not, then they should state why they think they’re different.*

Response

In light of the reviewer’s comments, it is clear that the taxonomic discussion of *Muntiacus vuquangensis/gigas* requires clarification from the outset. We agree that, following the study of Turvey et al. (2016), *Muntiacus vuquangensis* and *Muntiacus gigas* are synonyms, with *M. gigas* taking precedence. The title, keywords, abstract and manuscript has been updated accordingly and the taxonomic discussion moved to the introduction, as follows:

PAGE 2 lines 25-43.

Four years prior to the discovery and description of *M. vuquangensis*, however, Wei *et al.* [32] described a novel (and at the time, thought to be extinct) large muntjac species, *Muntiacus gigas*, based on examination of subfossil antlers from the Chinese Neolithic site of Hemudu in the Yangtze delta, dated 6 to 7 ka (“ka” = thousands of years before present). Recent work by Turvey *et al.* [7] on the morphometric characteristics of these specimens, with further samples from several Chinese sites ranging in date from the Late Pleistocene to Holocene (**figure 1; sites 1-8**), demonstrated that there are no morphological grounds to separate *M. gigas* specimens from extant *M. vuquangensis* and that these taxa should be considered conspecific. Given that the description of *M. gigas* predated the description of *M. vuquangensis*, the species name of the former has priority [7]. We therefore adopt the use of *M. gigas*, which we consider to be synonymous with *M. vuquangensis*.

1.3 Comment

*When comparing measurements between the new specimen and existing data for other muntjacs, the authors should also compare their specimen with available measurement data for *Muntiacus gigas* – this is really important to do.*

Response

We agree and we sort available comparative measurements for *M. gigas* before submission of the manuscript and made renewed searches and appeals to colleagues following the reviewer’s comments. We were successful in sourcing limited data for “*M. vuquangensis*” which as per the discussion above, is regarded as the same species. The original description of *M. gigas* fragmentary mandibles and maxilla are alluded to but no descriptions or measurements are presented (Wei et al. 1990). Subsequent records also appear to have focused on the identification and description of pedicles and antlers. To our knowledge there are no available published dental data with a confirmed ID for *M. gigas*: we are therefore grateful to R1 for supplying m2 and m3 length data for this taxon.

These data were incorporated as follows:

PAGE 7 lines 3-5

Lengths of the m2 and m3 (SLML = 12.58 mm and TLML = 16.71 mm, respectively) from a specimen of *M. gigas* (T301-6) from Tianluoshan, near Hemudu (Turvey, personal communication) also fall within the range of *M. muntjak*.

PAGE 7 lines 10-12

Comparison of m2 length (SLML) and m3 length (TLML) from the *M. gigas* specimen indicated no significant difference between these measures and the normative sample of *M. muntjak*.

PAGE 8 Lines 9-14

A comparison of Middle and Late Pleistocene m3 lengths from specimens attributed to “*M. muntjak*” with data from extant *M. muntjak* ($n = 32$; this study, [41]) does suggest larger size in the Pleistocene (figure 7): the difference in median values is statistically significant (Kruskal-Wallis: H_c (tie-corrected) = 43.03; $P = < 0.001$). Conversely, comparison of median values of Middle and Late Pleistocene records and *M. gigas* ($n = 4$; including HBC-27587) indicated no significant difference (Kruskal-Wallis: H_c (tie-corrected) = 0.2468; $P = 0.8839$).

See also response to comment below:

1.4 Comment

The lack of any statistical analysis to compare the new specimen with other muntjac samples (muntiacus, vuquangensis, gigas) is a major omission, and really has to be done – even if just via something fairly straightforward such as t-tests or ANOVAs. A statistical assessment of measurement differences, rather than just “eyeballing” the data, is essential.

Response

We agree that the study would benefit from statistical testing of observations of metric data. Given that the archaeological specimen is a single sample simple t-tests or ANOVA’s would not be appropriate, given $n = 1$, we have used a normative comparisons approach (described below) of a single case (dental metrics from the mandible) to a normative sample (metric data sets from comparative taxa).

PAGE 3 lines 55-60 to PAGE 4 lines 3-11

The **methodology** has been updated accordingly:

Statistical tests were performed to compare dental metrics of comparative museum specimens and published sources [41-43]. Comparison of dental metrics from the mandible, and in cases where sample sizes of comparative data were small, testing followed the procedures for normative comparisons [44-47], where metrics from a single case (e.g. HBC-27587) are compared with a normative sample generated from comparative data to test for differences from the estimated parameters of a given taxon (e.g. *M. muntjak*). Multivariate normative comparisons (m2 and m3, lengths and widths or a subset of these data depending on the availability of comparative data) were performed using a modified Hotelling’s T^2 test, following Huizenga *et al.* [46]. Univariate normative comparisons were also performed with a modified t -test [44-45] with a step-down correction to control familywise false-positive error rate in multiple comparisons [47]. These approaches assume univariate/multivariate normality in the normative samples. Sample data sets were examined for departures from normality using the functions in PAST 3.20. Normative comparisons were performed using the E-clip, Multivariate and Univariate Normative Comparisons online platform [48] accessed at: eclip.shinyapps.io/NormativeComparisons/.

The comparative section (**3.2 Morphology and metrics - comparisons**) has been revised in the following areas:

PAGE 5 lines 47-50 (comparison with *C. pygargus* – see also response to comments from R2 “*Capreolus*”, below)

The m2 and m3 lengths of the archaeological specimen exceed those of small samples of Holocene specimens from Primoye [43] but these differences are not significant (SLML: $n = 4$; $t = 1.117$; $P = 0.25$, one-sided; TLML: $n = 5$; $t = 1.291$; $P = 0.09$, one-sided).

PAGE 6 lines 20-22 (comparison with *E. cephalodus* – see also response to comments from R2 “*Elaphodus*”, below)

Comparison of m3 lengths and widths (TLML, TLMW) with a sample of extant *E. cephalophus* ($n = 9$; this study, [42]) indicate that these dimensions are significantly larger in the archaeological specimen ($T^2 = 26.498$; $P < 0.001$; one-sided).

PAGE 6 lines 54-60 and PAGE 7 lines 3-25

The limited available measurements from larger-bodied *Muntiacus* spp., *M. crinifrons* ($n = 2$) and *M. feae* ($n = 2$) fall within the upper range of *M. muntjak*. Multivariate normative comparisons of m2 and m3 lengths and widths (SLML, SLMW, TLML, TLMW) of available data from these species ($n = 2$, in both cases) with sample data from specimens of *M. muntjak* from mainland Southeast Asia ($n = 23$) indicate no significant difference (**table 3**).

Available data from two specimens of “*M. vuquangensis*”, the young male (NHMUK 2010.484) (NHMUK 2010) and the 34 kg female (AMNH M-274588), indicate that measurements from giant muntjac overlap with the upper range of *M. muntjak* (**figure 5**). Lengths of the m2 and m3 (SLML = 12.58 mm and TLML = 16.71 mm, respectively) from a specimen of *M. gigas* (T301-6) from Tianluoshan, near Hemudu (Turvey, personal communication) also fall within the range of *M. muntjak*. Multivariate normative comparisons of m2 and m3 lengths and widths (SLML, SLMW, TLML, TLMW) of the young male *M. vuquangensis* with sample data from *M. muntjak* ($n = 23$) indicated no significant difference (**table 3**). This test indicated that the metrics from Schaller’s female specimen are marginally, but significantly, larger than the normative sample (**table 3**). Comparison of m2 length (SLML) and m3 length (TLML) from the *M. gigas* specimen indicated no significant difference between these measures and the normative sample of *M. muntjak*.

The dimensions of the m2 and m3 of HBC-27587 exceed all available comparative data from species within the tribe (**figure 5**). Comparisons of m2 and m3 lengths and widths ($n = 23$) and m3 lengths and widths ($n = 32$) of HBC-27587 indicate that these variables are significantly larger in the archaeological specimen, than in the sample data for *M. muntjak* (**table 3**). The body weight estimates derived from measurements of the mandible are also consistent with the trend in dental metrics and suggest a larger animal than the majority of *Muntiacus* spp. A mean estimate of approximately 40 kg for the specimen exceeds reported weights ranges of all taxa within the tribe bar one: *M. gigas* (**figure 6**). While the dental dimensions of the Hang Boi specimen exceed available comparative data, given the range of variation within dental measurements in the genus (**figure 5**) there is no reason to suggest that the archaeological specimen would be an excessively or anomalously large specimen for this taxon. As such, there are no compelling grounds to posit a novel species and we refer specimen HBC-27587 to *M. gigas* (syn. *vuquangensis*).

Addition of a table (**table 3**):

Table 3. Multivariate (modified Hotelling’s T^2 test) and univariate (modified t -test) normative comparisons [46, 48] of dental metrics of *Muntiacus* spp. and HBC-27587 with samples of *M. muntjak*. Significant differences are shown in bold.

Multivariate normative comparisons: SLML, SLMW, TLML, TLMW.			
Normative sample: M. muntjak n = 23 [this study].			
Specimen	Hypothesis	T^2	P
M. vuquangensis (NHMUK 210.484)	One-sided (larger)	1.211	0.169
M. vuquangensis (AMNH M-274588)	One-sided (larger)	2.269	0.05
HBC-27587 (this study)	One-sided (larger)	2.468	0.04
M. feae (AMNH 32.11.1.17.1)	Two-sided (difference)	0.642	0.639
M. feae (AMNH 24.1.6.2)	Two-sided (difference)	1.888	0.154
M. crinifrons (AMNH M-56991)	Two-sided (difference)	0.859	0.506
M. crinifrons (NHMUK 1.3.2.21)	Two-sided (difference)	0.985	0.439
Multivariate normative comparisons: TLML, TLMW.			
Normative sample: M. muntjak n = 32 [this study, 41].			
Specimen	Hypothesis	T^2	P
HBC-27587 (this study)	One-sided (larger)	4.966	0.007
Univariate normative comparisons: SLML and TLML.			
Normative samples: SLML M. muntjak n = 23 [this study] and TLML M. muntjak (n = 32 [this study, 41].			
Specimen	Hypothesis	t	P
SLML M. gigas (T301-6)	One-sided (larger)	0.497	0.318
TLML M. gigas (T301-6)	One-sided (larger)	0.266	0.394

Minor comments:

- “sub-fossil” should be changed throughout the manuscript to “subfossil” (the more normal way of spelling this word) **Changed to “subfossil” throughout the manuscript**

- “critically endangered” should be capitalised to “Critically Endangered”, as this is a formal IUCN threat category. It also shouldn’t be written in inverted commas (line 58) **Changed to Critically Endangered, throughout.**

- Line 112: “11, 778” – spacing after the comma needs correcting **Corrected**

- Line 243: “ectosylids” spelled incorrectly **Corrected**

- Line 257: “Muntiacini” shouldn’t be italicised **Corrected**

- Line 263: “celaphodus” spelled incorrectly **Corrected**

- Line 285: not strictly true that the only confirmed gigas specimens derive from Holocene sites in China - Turvey et al 2016 also included a Late Pleistocene specimen from Yuhang in their analyses, so Pleistocene material has been interpreted by previous authors as representing gigas

Agreed, this statement in the original manuscript was not correct.

Figure 1 and caption have been revised to show Yuhang (site 8) as a Late Pleistocene site. The following sections have also been revised:

PAGE 2 lines 29-31

Recent work by Turvey *et al.* [7] on the morphometric characteristics of these specimens, with further samples from several Chinese sites ranging in date from the Late Pleistocene to Holocene (**figure 1; sites 1-8**),

PAGE 7 lines 32-34

At present, however, the only confirmed giant muntjac fossils derive from one Late Pleistocene site (Yuhang; **figure 1; site 8**) and a handful of Holocene archaeological sites in China, collectively dating between 2.2 and 7 ka [7].

- Line 352-3: *even if lower elevational estimates are confounded by human activity, it would still be useful to say what the current lower elevational range estimate is for the species, and how much this differs from the elevation of the Holocene site from which the new specimen was found - and also discuss the idea of the Annamites acting as an altitudinal refugium for the species, in the context of this concept more widely for SE Asian and other mammal species*

This section is revised and expanded:

PAGE 8 lines 36-43

The species has been recorded up to 1200 m above sea level (asl) but available data suggests the species is most likely to occur below 1000 m asl. Reliable estimates of lower elevational limits, however, are confounded by recent trends in habitat removal and over-hunting [31]. The Trảng An karst rarely approaches 200 m asl and the coastal plain that surrounds the massif, and dolines within it, are only a few meters above sea level. The specimen from Hang Bôi indicates that giant muntjac occurred in a lowland karst setting and, as such, the current altitudinal range of the giant muntjac in the Annamites may not represent an ecological preference for habitats at higher altitudes, but rather a response to human activities.

REVIEWER 2

Dear authors,

this is a carefully conducted transdisciplinary study, a very well written manuscript, and an excellent example that we need the big picture, far beyond historical time, when assessing conservational needs. Congratulations, I've enjoyed reading very much! I've only indicated minor suggestions to consider in the manuscript. The two most important among them, I'm coming up with here again:

2.1 Comment

*Critical data of *Capreolus pygargus* and *Elaphodus cephalophus megalodon* seem to have been neglected. In order to make the issues more transparent I'd suggest to add relevant data.*

Response

Data have been added to the treatment of *Elaphodus* and *Capreolus* and these sections have been revised as per the reviewer's comments:

Capreolus:

PAGE 5 lines 39-60 and PAGE 6 Lines 3-4

While extant populations of Siberian roe deer, *Capreolus pygargus*, have a largely temperate distribution and currently range only as far south as Eastern Tibet and central Chinese provinces [57], these are mid-sized cervids (32-48 kg [58]) with similar tooth dimensions and

simple occlusal morphology. Average measurements of m2 lengths and widths (SLML, SLMW) from late Pleistocene samples of “*C. manchuricus*”, regarded to be synonymous with *C. pygargus*, from Jilin Province, Northeast China [59] and reference data (NHMUK 83.324) suggest similar dimensions to HBC-27587 (**figure 5**). Lengths and widths of the third molar are smaller than the archaeological specimen and fall into the upper size range of equivalent data for *M. muntjak* (**figure 5**). The m2 and m3 lengths of the archaeological specimen exceed those of small samples of Holocene specimens from Primoye [43] but these differences are not significant (SLML: $n = 4$; $t = 1.117$; $P = 0.25$, one-sided; TLML: $n = 5$; $t = 1.291$; $P = 0.09$, one-sided).

Morphologically, the concavity of the posterior margin of the mandibular ramus appears shallower but much wider (i.e. extends much further dorsally to the condyle) in *Capreolus* than is suggested in the archaeological specimen. *C. pygargus* lacks anterior cingulids, as is the case with the archaeological specimen, but protoconids and hypoconids are much more angular and there is marked overlap of the metaconids and entoconids. Ectostylids are present on the lower molars, though are weakly developed, particularly on the m3, in comparison to HBC-27587 (**figure 3**).

The genus *Capreolus* is associated with temperature climates at higher latitudes both in terms of extant populations and the wider fossil record in Eurasia [60]. In Pleistocene records of East Asia, *C. pygargus* does not appear as a component of the *Ailuropoda-Stegodon* fauna [61] and Holocene records (archaeological and historical, $n = 51$) are restricted to central and northern provinces in China, with no records of the species further south than Hubei [11]. As such, *C. pygargus* can also be reasonably discounted on biostratigraphic and as well as ecological grounds (**see section 4.2**).

Limited available data from *Capreolus pygargus* is also included in **figure 5** and occlusal morphology of the m2 and m3 is now shown in **figure 3**.

Elaphodus:

PAGE 6 lines 12-22

The tufted deer, *Elaphodus cephalophus*, is currently restricted to southern China (with historical records from eastern Myanmar). These deer are associated with montane forest habitats and apparently do not range into sub-tropical environments [63]. Hooijer [62] describes a larger, Pleistocene subspecies, *E. cephalophus megalodon*, from Yanjinggou in China. Metrics of individual teeth are not reported, but an upper M1 to M3 length of 410 mm suggest tooth dimensions may be similar to, if not exceeding those of larger *Muntiacus* spp. [cf. 25]. This taxon, however, is known only from early Middle Pleistocene sites in China [64]. Conversely, reported body weight ranges (17 – 30 kg; **figure 6**) and metrics of the m2 and m3 (**figure 5**) indicate that extant *Elaphodus* are relatively small in comparison to equivalent data from the archaeological specimen. Comparison of m3 lengths and widths (TLML, TLMW) with a sample of extant *E. cephalophus* ($n = 9$; this study, [42]) indicate that these dimensions are significantly larger in the archaeological specimen ($T^2 = 26.498$; $P < 0.001$; one-sided).

REVISIONS FOLLOWING ANNOTATED PDF

PAGE 2 lines 15-17

Line 49 (63) “short” changed to “shorter”

Line 50 (64) “thick” changed to “thicker”

Line 50 (64) “large” changed to “larger”

PAGE 3 lines 39-40

Comment “Which institution? The information is important for future research.”

Clarified: “The specimen is to be stored and curated by the Trảng An Management Board, Ninh Binh.”

PAGE 4 line 35

Comment “the term “molar” includes that the tooth is permanent, hence “permanent” can be deleted here”

“permanent” deleted

PAGE 4 lines 35-36

Comment “I'd recommend to specify more here and say “in medium wear” as the state of wear is significant for the interspecific occlusal outline.”

Agreed, wear stage is critical here and was dealt with in rather vague terms in the original manuscript. Here

wear stage is standardised and described, following:

[40] Anders U, von Koenigswald W, Ruf I, Smith BH. 2011 Generalized individual dental age stages for fossil and extant placental mammals. *PalZ* 85, 321-39.

Revised to:

“The second (m2) and third (m3) lower molars are in situ and in wear (equivalent to IDAS late stage 3 or 4), indicating an adult animal (**figure 4**).”

The caption to **figure 3** is updated:

Figure 3. Annotated characters of the mandible (top) and second and third lower molars (bottom) referred to in the text. Molars of two species of Bovidae (*Pseudoryx nghetinhensis* and *Capricornis [sumatraensis] maritimus*) and four species of the Cervidae (*Muntiacus vuquangensis*, *Hyelaphus porcinus*, *Hydropotes inermis* and *Capreolus pygargus*) are shown to scale with specimen HBC-27587. All specimens are equivalent to individual dental age stages (IDAS) late stage 3/stage 4 [40].

The methodology is updated (**PAGE 3 lines 53-54**)

Wherever possible, reference specimens with equivalent individual dental age stages (IDAS) [40] to the archaeological specimen (IDAS 3 to 4) were prioritised for morphological comparisons of the dentition.

Comment “What do you mean with “molars appear rounded”? The occlusal outline (what were not less rounded at the same level, when wear were less)? I'd suggest to rethink, if this part of the sentence makes sense.”

Agreed, this sentence is not clear and is redundant given the proceeding description. Deleted: “Wear to the occlusal surfaces is marked and the molars appear rounded.”

Comment ““cuspid” in lower teeth”

Comment “Anterior cusp? Should be “cuspid” (see comment above”, but which one: metaconid or protoconid?”

Comment “which one: hypoconid or entoconid? “cuspid” in lower teeth”

Comment posterior cusp changed to “entoconid”

This section is revised for clarify and to use more specific terminology (**PAGE 4 lines 37-42**):

The labial margins of the teeth are intact. On the m2, a worn but robust ectostylid is located between the protoconid and hypoconid. The anterior margin of the m2 has lost enamel at the point of contact with the posterior margin of the m1 (not present). No anterior cingulum is

evident. The posterior margin of the m2 has lost enamel at the contact point with the anterior margin of the m3. Enamel is present on the labial margin of the hypoconid, though the entoconid is broken at the location of the metastylid. Enamel has broken away along the remainder of the lingual edge of the metaconid.

PAGE 4 line 50

“No back fossa is evident, though the hypoconulid is broken and this character is lost through wear.” changed to “The back fossa is lost through wear.” as per R2’s suggestion.

Comment “What is “figure 3” referring to? It doesn’t show *Hydropotes* nor the term “goat folds”.

This issue was also flagged by R3 and was not clear in the original manuscript.

Figure 3 is now updated to include illustration of *Hydropotes* occlusal surface – anterior cingulids (“goat folds”) are annotated. The term “goat folds” is dropped and the formal terminology (anterior cingulids) is used:

(PAGE 5 lines 24-25)

“appear to have a more complex molar morphology with anterior cingulids on the lower molars (**figure 3**).”

*“I’d say this is a critical species when trying to identify the archaeological specimen. To give your decision for *M. vuquangensis* more ground you should include data in tables, diagrams and figures. My personal note on *C. pygargus* say that it is a bit more higher crowned than *Muntiacus*.”*

See response “*Capreolus*”, above. We have included *Capreolus pygargus* in **figure 3**, comparative data in **Table S2** and updates to **figures 3** and **5**, accordingly.

Comment “the third lobe of lower m3 in ruminants is intraspecifically pretty variable, hence you should have consulted a large sample to judge on this”

Agreed, this is a variable character and not confirmed in our analysis. Reference material for this material was reviewed: see revision for “*Capreolus*”, above.

PAGE 6 line 19

Line 227 “*Elaphus*” changed to “*Elaphodus*”

Comment “Do tooth metrics coincide with your archaeological specimen? This is important information, which should not be neglected”

See response to comment concerning “*Elaphodus*”, above.

Comment “above you’ve chosen to say Lao PDR; I’d suggest to stay with one name”

PAGE 6 line 42-43 “Laos” changed to “Lao PDR”

Comment “*Hassanin et al. 2012* and *Heckeberg et al. 2016* suggest that *M. reevesi* and *M. vuquangensis* had a common ancestor.”

PAGE 7 line 31

Revised to “Analyses of mitochondrial DNA suggest that giant muntjacs appeared in the Early Pleistocene between 0.9 to 1.8 million years ago [27].”

Comment “verb missing?”

PAGE 7 lines 50-51

“This likely to be a complex issue to address at present and a matter of interpretation.” corrected to “This is likely to be a complex issue to address at present and a matter of interpretation.”

Comment “*find this sentence hard to understand, perhaps separating into two sentences would help.*”

This section has been revised and clarified. Please also note that on review, the original “*M. muntjak*” data set, erroneously contained data from *M. feae* and *M. crinifrons*. While it is unlikely that these small samples had a significant impact on the characteristics of the distribution of measurements (as discussed in the manuscript in the morphometric description of these species), it was not appropriate to include them and the Kruskal-Wallis tests were re-run. The changes to the outcome of these tests were marginal and subsequent interpretation was not affected. Furthermore, the data point for m3 length for *M. gigas* (kindly supplied by R1) was grouped with m3 lengths of *M. vuquangensis* (synonymous) and HBC-27587 (the archaeological mandible) as a sample for *M. gigas*, to compare with the Middle and Late Pleistocene data sets.

PAGE 8 lines 9-19

A comparison of Middle and Late Pleistocene m3 lengths from specimens attributed to “*M. muntjak*” with data from extant *M. muntjak* ($n = 32$; this study, [41]) does suggest larger size in the Pleistocene (**figure 7**): the difference in median values is statistically significant (Kruskal-Wallis: H_c (tie-corrected) = 43.03; $P < 0.001$). Conversely, comparison of median values of Middle and Late Pleistocene records and *M. gigas* ($n = 4$; including HBC-27587) indicated no significant difference (Kruskal-Wallis: H_c (tie-corrected) = 0.2468; $P = 0.8839$).

In the Pleistocene data sets, m3 length does not appear to vary as a function of time between Middle and Late Pleistocene (Mann Whitney $U = 702$; $P = 0.7373$), although sample sizes are uneven (Late Pleistocene: $n = 99$; Middle Pleistocene: $n = 15$). Median values also do not appear to differ as a function of latitude (Late Pleistocene Vietnam, $n = 30$; Late Pleistocene China, $n = 69$; Mann Whitney $U = 786$; $p = 0.0597$).

Comment “*I suggest to put in parentheses for the sake of consistency*”

PAGE 9 lines 29-30 “and civet, *Paradoxurus hermaphroditus*.” changed to “and civet (*Paradoxurus hermaphroditus*).”

COMMENTS ON FIGURE 2

“*is that meters above sea level?*”

“*this figure shows less layers than given in Figure S1. Specification is needed.*”

Figure 2 shows the upper 2 m of the stratigraphic section and heights relative to an arbitrary site datum (set at 500 m) rather than sea level. This issues are clarified in revised figure caption:

Figure 2. Hang Boi. (A) Plan of cave showing excavation area: grid square 226/109 and section are indicated (B) looking NNW/NW during excavation through shell midden, and (C) upper 2 m of stratigraphic section (east facing) of grid square 226/109 showing calibrated radiocarbon dates from charcoal and layer (5105) containing specimen HBC-27587. Heights shown are relative to arbitrary site datum set at 500m.

The caption for **table S1** is also amended for clarity.

Table S1. AMS radiocarbon dates from the total excavated extent of the midden sequence from Hang Boi (HBC), Tràng An World Heritage Area, Northern Vietnam.

REVIEWER 3

The manuscript is well written and if the topic falls into the scope of the journal, it definitely is worth to be published.

3.1 Comment

Unfortunately I have not found any access to the supplementary tables and figures. Therefore, I could not comment some of the evidence used in the ms.

Response

We regret that the R3 could not gain access to the supplementary information, as we feel the contents address several of the reviewer's comments, below:

3.2 Comment

Substantial part of the arguments is based on the body mass estimation. The method is recalled from Janis (1990). Since the reference deals with a chapter from a book published almost 30 years ago, it may be difficult for many readers to get and read the methodology there. This may have two implications to the recent manuscript. First, the authors of the present study should describe in brief the principles of the method for those readers who do not have the cited book at hand.

Response

In light of the reviewer's methodological concerns that potential readers may not have access to the source material (Janis, 1990) and for the sake of transparency, the parameters of the regression equations and output (which were originally included in the in Supplementary Materials as table S2) have been included as table 2 in the revised manuscript. The methodology section is also revised:

PAGE 3 lines 43-47 Body mass estimates were derived from six measurements of the specimen, following the least squares regression equations and percentage standard errors of estimate (%SEE) for the Cervidae of Janis [39; table 2].

3.3 Comment

Second, from a more recent literature (e. g., Mendoza, M., Janis, C. M. & Palmqvist, P., 2006. Estimating the body mass of extinct ungulates: a study on the use of multiple regression. J. Zool. 270, 90-101; De Esteban-Trivigno, S. & Köhler, M., 2011. New equations for body mass estimation in bovids: Testing some procedures when constructing regression functions. Mamm. Biol. 76, 755-761) it is apparent that some improvements of the method are available since 1990. Even if it was not applicable to the paleo-material available, it seems the body mass estimation based on multiple regression should be considered more carefully.

Response

While the methods, above, present refined means of estimating body mass, they are either practically (deal with characters or character sets not available to this study) or taxonomically (Bovidae, not Cervidae) inappropriate. To our knowledge, the equations of Janis (1990) remain the most comprehensive and robust treatment of body mass estimations for Cervidae (see also Suraprasit et al., 2016).

Suraprasit K, Jaeger JJ, Chaimanee Y, Chavasseau O, Yamee C, Tian P, Panha, S. 2016 The Middle Pleistocene vertebrate fauna from Khok Sung (Nakhon Ratchasima, Thailand): biochronological and paleobiogeographical implications. *ZooKeys* **613**, 1.

3.4 Comment

Ad Table 1 – These are important measures. A sketch showing simply how the mandible was measured would help.

Response

A figure showing these measurements was (and is) included in the Supplementary Information as **figure S1**.

3.5 Comment

Ad Figure 3 – Chinese water deer announced in the text (lines 196-198) is absent in the figure.

Response

Hydropotes now shown in **figure 3** (see also response to **R2** comments, above)

ANNOTATED PDF - COMMENTS ON SUPPLEMENTARY INFORMATION

Table S1

Figure 2 says “5105”?

Correct. As discussed in **section 2.2** of the manuscript (**PAGE 3 lines 17-18**): “The mandible was recovered in the western half of a 1 m x 1 m grid square (226/109) in a 10cm unit of excavation (5010) through context (5105)”. Table S1 gives the excavation unit 5010 (within which the charcoal sample was recovered) rather than the context (5105): this is corrected to 5105 in the table, for clarity.

Comment. “*Why not using here the same abbreviations as for Table S2 and Figure S1? Otherwise you have to explain that L and M are equal with SLML, SLMW, TLML, and TLMW*”

Revised:

Column headings changed from “m2 L”, “m2 W”, “m3 L” and “m3 W” to “SLML”, “SLMW”, “TLML” and “TLMW”, respectively.

Appendix B

RESPONSE TO REFEREES (post revision)

Dear Andrew,

We would like to thank you, the editors and the referee for reconsidering our submission “*An 11,000-year-old giant muntjac subfossil from Northern Vietnam: implications for past and present populations*”. We are delighted that the manuscript is accepted on condition of minor revisions. These issues have been addressed and we hope that the manuscript is to the journals’ satisfaction.

We would like to express our sincere thanks for your assistance and diligent handling of queries during the revision process. The extra time taken by the referee to reconsider the manuscript is much appreciated. We are pleased that the revised submission has addressed the original comments and suggestions for improvement: we feel that the manuscript benefitted greatly for their input. We are also grateful for their critical eye and for highlighting the necessary minor revisions in the revised submission, which we address below.

Reviewer comments to Author:

Comments to the Author(s)

The authors have done a great job in addressing all of my comments and suggestions, and this is a great paper - I look forward to it being published! The only very minor further edits that I'd suggest are a few typos here and there:

- Page 3, lines 56-58: *It feels like there's a word missing from this sentence - maybe it needs a "For" at the start of the sentence?* Amended to “For comparison”
- Page 5, line 47: *should "Primoye" be "Primorye"?* Corrected to “Primorye”
- Page 5, line 57: *"temperature" should be "temperate"* Corrected to “temperate”
- Page 6, line 4: *remove "and" after "biostratigraphic"* Deleted: “and”
- Page 6, line 50: *should "(NHMUK 2010)" be removed? (not sure what this means after the actual reported specimen number)* Correct, this is a typo. Deleted: “(NHMUK 2010)”
- Page 7, line 49: *remove "of" from before "are present in Quaternary archives".* Deleted: “of”
- Page 8, line 3: *remove "of" after "published metric identifications".* Deleted: “of”
- Page 8, line 6: *"post glacial" is usually either hyphenated or one word.* Amended to: “post-glacial”
- Figure 3: *"Hydropotes" is spelled wrong in the figure, and "Capreolus (capreolus) pygargus" should have the subgenus name in parentheses removed (since it's incorrectly capitalised, and you don't refer to this subgenus name in the main text).* Agreed, many thanks for spotting these errors. Figure 3 has been amended: *Hydropetes* corrected to “*Hydropotes*”. The caption for *Capreolus* revised to “*Capreolus pygargus*”.